# ALGEBRAIC SPD AND CORRELATION GEOMETRY: A GYRO APPROACH

## ABSTRACT

The generalization of Deep Neural Networks (DNNs) to Riemannian manifolds has garnered significant attention across various scientific fields. Recent studies have demonstrated that several manifolds, including hyperbolic, spherical, Symmetric Positive Definite (SPD), and Grassmann manifolds, admit gyro-structures—powerful algebraic structures that enable the principled extension of DNNs to manifolds. Inspired by these advancements, we introduce a novel gyro-structure for SPD manifolds, leveraging the flexible and powerful Power-Euclidean (PE) geometry. Moreover, full-rank correlation matrices, which are scale-invariant, serve as compact representations of SPD manifolds. Consequently, we propose two novel gyro-structures for correlation matrix manifolds, based on two theoretically and empirically convenient metrics: Euclidean-Cholesky (EC) and log-Euclidean-Cholesky (LEC) geometries. Extensive experiments on knowledge graph completion tasks validate the effectiveness of our proposed gyro-structures.

## 1 INTRODUCTION

Deep Neural Networks (DNNs) have driven significant progress across numerous areas (Krizhevsky et al., 2012; He et al., 2016; Vaswani et al., 2017). Typically, DNNs operate under the assumption that data conform with Euclidean geometry. However, in many scientific fields, data possess a strongly non-Euclidean latent structure, such as Riemannian manifolds (Bronstein et al., 2017). Therefore, substantial efforts have been made to extend DNNs to manifolds (Huang & Van Gool, 2017; Huang et al., 2018; Brooks et al., 2019; Nguyen et al., 2019; Wang et al., 2021; Nguyen, 2021; Chen et al., 2023; 2024a; Wang et al., 2024b;a). Recently, several manifolds has been proven to admit gyrovector structures (Nguyen, 2022; Ungar, 2005), which naturally extends the Euclidean vector structure. Leveraging gyro-structures, several typical DNNs have been generalized into different geometries in a principled manner. Such manifolds include matrix manifolds, such as Symmetric Positive Definite (SPD) (Nguyen, 2022; Nguyen & Yang, 2023), Grassmann (Nguyen, 2022), Symmetric Positive Semi-definite (SPSD) manifolds (Nguyen & Yang, 2023), and constant curvature spaces, such as hyperbolic (Ganea et al., 2018; Shimizu et al., 2020) and spherical (Skopek et al., 2019) manifolds.

As shown by Cruceru et al. (2021); Nguyen et al. (2024), matrix manifolds such as the SPD manifold, provide a compelling balance between structural richness and computational feasibility, serving as appealing alternatives to hyperbolic spaces. Nguyen (2022) identified three gyro-structures over the SPD manifold based on Affine-Invariant (AI) (Pennec et al., 2006), Log-Euclidean (LE) (Arsigny et al., 2005), and Log-Cholesky (LC) (Lin, 2019) metrics. Apart from the above metrics, Power-Euclidean (PE) metric (Dryden et al., 2010) has also shown promising performance in different applications Li et al. (2017); Wang et al. (2020); Chen et al. (2024c;b). However, the gyro-structure under PE geometry remains unexplored.

On the other hand, the manifold of full-rank correlation matrices has recently garnered increasing attention, as it provides a normalized, compact representation of the SPD manifold (David & Gu, 2019; Thanwerdas & Pennec, 2022b). Since full-rank correlation matrices are scale-invariant, they are particularly effective for representing data where scale is irrelevant to the problem (Thanwerdas, 2024). In domains such as Diffusion Tensor Imaging (DTI) (Pennec et al., 2006), Brain-Computer Interfaces (BCI) (Jalili & Knyazeva, 2011; Barachant et al., 2013), and Gaussian graphical networks (Epskamp & Fried, 2018), using correlation matrices instead of covariance matrices is both natural and effective. Recently, several Riemannian structures have been proposed for full-rank correla-

tion matrices, including the Euclidean-Cholesky (EC) and Log-Euclidean-Cholesky (LEC) metrics (Thanwerdas & Pennec, 2022b). This motivates us to further explore algebraic gyro-structures for full-rank correlation matrices.

Based on the above discussions, we propose a novel SPD gyro-structure based on PE geometry alongside two new gyro-structures for full-rank correlation matrices induced by EC and LEC geometries, respectively. On the SPD manifold, as the PE metric converges to the LE metric when the matrix power approaches zero, the proposed PE gyrovector structure naturally recovers LE gyro space. We expect the proposed PE gyro-structure to provide a flexible alternative to the existing LE gyro space. Regarding correlation matrices, we emphasize that these two proposed gyro-structures are the first to be introduced for this manifold. We anticipate that these algebraic structures will advance deep learning over correlation matrices. In summary, our **main contributions** are summarized as follows:

- **SPD gyro-structure based on PE geometry.** We propose a novel SPD gyro-structure under the PE geometry, which recovers the LE gyro when power tends to 0.

- **Two novel correlation gyrovector structures.** Inspired by correlation matrices as the compact alternatives of the vanilla covariance matrices, we propose two novel gyro-structures for full-rank correlation matrices, induced by the theoretically and computationally efficient EC and LEC metrics.

- **Empirical validation in knowledge graph completion tasks.** We validate the effectiveness of our approach through extensive experiments on knowledge graph completion tasks, demonstrating the effectiveness of our algebraic gyro structures.

**Main theoretical results.** Lem. 3.1 and Lem. 3.2 introduce the binary operation and scalar multiplication under SPD manifolds based on the PE geometry, respectively. We then define the SPD gyro-structure under the PE geometry in Thm. 3.3. Additionally, we propose two new gyro-structures for full-rank correlation matrices induced by EC and LEC geometries. The corresponding binary operation and scalar multiplication are provided in Lem. 5.1, Lem. 5.4, Lem. 5.4 and Lem. 5.5. The definitions of gyrovector space are presented in Thm. 5.3 and Thm. 5.6. Due to page limits, all the proofs are placed in App. C.

## 2 PRELIMINARIES

### 2.1 GYROVECTOR SPACES

Gyrovector spaces naturally extend the vector structures into manifolds (Ungar, 2005; 2014; 2022; Nguyen, 2022). We briefly review gyrogroups and gyrovector spaces in the following. For more in-depth discussions, please refer to Ungar (2005; 2012; 2014).

**Definition 2.1** (Gyrogroups (Ungar, 2014)). A pair $(G, \oplus)$ is a groupoid in the sense that it is a nonempty set, $G$, with a binary operation, $\oplus$. A groupoid $(G, \oplus)$ is a gyrogroup if its binary operation satisfies the following axioms for $a, b, c \in G$:

(G1) There is at least one element $e \in G$ called a left identity such that $e \oplus a = a$.

(G2) There is an element $\ominus a \in G$ called a left inverse of $a$ such that $\ominus a \oplus a = e$.

(G3) There is an automorphism $\mathrm{gyr}[a, b] : G \to G$ for each $a, b \in G$ such that

$$a \oplus (b \oplus c) = (a \oplus b) \oplus \mathrm{gyr}[a, b]c \quad \text{(Left Gyroassociative Law)}. \tag{1}$$

The automorphism $\mathrm{gyr}[a, b]$ is called the gyroautomorphism, or the gyration of $G$ generated by $a, b$.

(G4) $\mathrm{gyr}[a, b] = \mathrm{gyr}[a \oplus b, b]$ (Left Reduction Property).

**Definition 2.2** (Gyrocommutative Gyrogroups (Ungar, 2014)). A gyrogroup $(G, \oplus)$ is gyrocommutative if it satisfies

$$a \oplus b = \mathrm{gyr}[a, b](b \oplus a) \quad \text{(Gyrocommutative Law)}. \tag{2}$$

Building on this foundation, some researchers expanded the concept (Kim, 2015; 2016; 2020a;b). The work in Nguyen (2022) proposed a more rigorous definition of gyrovector spaces and key operations on them.

**Definition 2.3** (Gyrovector Spaces (Nguyen, 2022)). A gyrocommutative gyrogroup $(G, \oplus)$ equipped with a scalar multiplication

$$(t, x) \rightarrow t \odot x : \mathbb{R} \times G \rightarrow G \tag{3}$$

is called a gyrovector space if it satisfies the following axioms for $s, t \in \mathbb{R}$ and $a, b, c \in G$:

(V1) $1 \odot a = a, 0 \odot a = t \odot e = e$, and $(-1) \odot a = \ominus a$.

(V2) $(s + t) \odot a = s \odot a \oplus t \odot a$.

(V3) $(st) \odot a = s \odot (t \odot a)$.

(V4) $\mathrm{gyr}[a, b](t \odot c) = t \odot \mathrm{gyr}[a, b]c$.

(V5) $\mathrm{gyr}[s \odot a, t \odot a] = I_d$, where $I_d$ is the identity map.

Given a manifold $\mathcal{M}$, the gyro operations can be defined by the following definition.

**Definition 2.4** (Gyro Operations (Nguyen, 2022)). Let $P, Q, R \in \mathcal{M}$, $t \in \mathbb{R}$ and $I$ denotes the identity element in $\mathcal{M}$, the gyro operations, such as binary operation, scalar multiplication and gyroautomorphism, are defined as:

$$P \oplus Q = \mathrm{Exp}_P \left( \Gamma_{I \rightarrow P} \left( \mathrm{Log}_I (Q) \right) \right), \tag{4}$$

$$t \otimes P = \mathrm{Exp}_I (t \mathrm{Log}_I(P)), \tag{5}$$

$$\mathrm{gyr}[P, Q]R = \left( \ominus (P \oplus Q) \right) \oplus \left( P \oplus (Q \oplus R) \right). \tag{6}$$

If a groupoid $(G, \oplus)$ conforms with axioms of Defs. 2.1 and 2.2, it forms a gyrocommutative gyrogroup. When endowed with the scalar multiplication $\otimes$, and if $(G, \oplus, \otimes)$ satisfies with axioms of Def. 2.3, it further forms a gyrovector space.

## 2.2 GEOMETRIES OF SPD AND FULL-RANK CORRELATION MATRICES

**The SPD geometry:** The space of SPD matrices forms a manifold, known as the SPD manifold (Arsigny et al., 2005), which has been successfully applied in various fields (Huang & Van Gool, 2017; Brooks et al., 2019; Sukthanker et al., 2020; Nguyen, 2022). To respect the non-Euclidean geometry, several Riemannian structures on the SPD manifold were proposed (Pennec et al., 2006; Arsigny et al., 2005; Lin, 2019; Bhatia, 2009; Thanwerdas & Pennec, 2022a). Due to the fast computation speed and theoretical convenience of the Power-Euclidean (PE) metric, and when the power tends to 0, this metric approaches the Log-Euclidean (LE) metric, building a bridge between Euclidean and LE metrics. Based on the above advantages, the PE metric has already seen successful applications in other fields (Zhou & Müller, 2022; Pennec, 2020; Pereira et al., 2024).

**The correlation geometry:** Full-rank correlation matrices, known as the open elliptope, have recently been described as a quotient manifold of SPD matrices due to the smooth, proper, and free congruence action of positive diagonal matrices (David & Gu, 2019). Specifically, a full-rank correlation matrix is obtained by dividing the covariance matrix by the standard deviation of each variable. Some applications can benefit from the well-suited geometric structure of this manifold, such as Brain Connectomes (Varoquaux et al., 2010), Gaussian graphical networks (Epskamp & Fried, 2018) and Phylogenetic trees (Garba et al., 2021). However, its geometry has been much less studied than that of SPD matrices. Recently, some researchers proposed some metrics based on full-rank correlation matrices: Euclidean-Cholesky (EC) and log-Euclidean-Cholesky (LEC) metrics (Thanwerdas & Pennec, 2022b). The basic operators based on different metrics are summarized in App. B.

## 2.3 NOTATION

The homogeneous Riemannian manifold is denoted as $\mathcal{M}$, $T_P \mathcal{M}$ is the tangent space at $P \in \mathcal{M}$ and $g_P(\cdot, \cdot)$ is the Riemannian metric at $P \in \mathcal{M}$. $\mathrm{Log}_P(\cdot)$ and $\mathrm{Exp}_P(\cdot)$ as the Riemannian logarithm and Riemannian exponential at $P$, $\exp(P)$ and $\log(P)$ as the usual matrix exponential and logarithm of $P$, $\Gamma_{P \rightarrow Q}(W)$ as the parallel transport of a tangent vector $W \in T_P \mathcal{M}$ from $P$ to $Q \in \mathcal{M}$ along geodesics connecting $P$ and $Q$, $d(\cdot, \cdot)$ as the geodesic distance, respectively. Denote by $\mathrm{M}_{n,m}$ the space of $n \times m$ matrices, $\mathrm{Sym}_n^+$ the space of $n \times n$ SPD matrices, $\mathrm{Sym}_n$ the space of $n \times n$ symmetric matrices, $\mathrm{Gr}_{n,p}$ the p-dimensional subspaces of $\mathbb{R}^n$, $\mathrm{Cor}_n^+$ the space of $n \times n$ full-rank correlation matrices, $\mathrm{LT}_n^0$ the space of the lower triangular part of $n \times n$ matrices with null diagonal

elements, $\mathrm{LT}_n^1$ the space of the lower triangular part of $n \times n$ matrices with unit diagonal elements, $\mathrm{Hol}_n$ the space of $n \times n$ symmetric matrices which diagonal elements are 0. Other notations will be introduced in appropriate paragraphs. Our notations are summarized in App. A.

## 3 PE GYROVECTOR SPACES OVER SPD MATRICES

This section investigates the gyro-structure of SPD manifolds based on PE geometry and uncovers the hidden connection between SPD manifolds in PE geometry and Euclidean space. As shown by Dryden et al. (2010), the PE metric recovers the LE metric with the matrix power approaching zero. In this sense, PE geometry can be viewed as a balanced metric of the vanilla LE metric. Furthermore, this metric enjoys theoretical and computational convenience. As shown by App. B, the associated Riemannian operators, such as exponential & logarithm maps and parallel transport, have simple closed-formed expressions. This motivates us to explore the gyro-structure of SPD manifolds with the power-Euclidean (PE) geometry. As the PE exponential map at the identity matrix $\mathrm{Exp}I(\cdot)$ is defined locally, the PE gyro addition and scalar product are well-defined only if the involved exponential map is well-defined.

**Lemma 3.1.** [↓] *For $P, Q \in \mathrm{Sym}_n^+$, the binary operation $P \oplus_{pe} Q$ is given as*

$$P \oplus_{pe} Q = (P^\alpha + Q^\alpha - I_n)^{\frac{1}{\alpha}}, P^\alpha + Q^\alpha - I_n \in \mathrm{Sym}_n^+, \tag{7}$$

*where $I_n$ is the $n \times n$ identity matrix.*

$I_n$ is the identity element of $\mathrm{Sym}_n^+$. Thus, using Eq. (7), we can obtain the inverse of P which is given by

$$\ominus_{pe} P = (2I_n - P^\alpha)^{\frac{1}{\alpha}}, 2I_n - P^\alpha \in \mathrm{Sym}_n^+. \tag{8}$$

**Lemma 3.2.** [↓] *For $P \in \mathrm{Sym}_n^+$ and $t \in \mathbb{R}$, the scalar multiplication $t \otimes_{pe} P$ is given as*

$$t \otimes_{pe} P = (tP^\alpha + (1-t) I_n)^{\frac{1}{\alpha}}, tP^\alpha + (1-t) I_n \in \mathrm{Sym}_n^+. \tag{9}$$

Similar to the gyrovector spaces on the Grassmann manifold (Nguyen, 2022), in the following, we implicitly assume the PE gyro operations are well-defined.

**Theorem 3.3.** [↓] $(\mathrm{Sym}_n^+, \oplus_{pe})$ *forms a gyrocommutative gyrogroup. Endowed with the scalar multiplication $\otimes_{pe}$, $(\mathrm{Sym}_n^+, \oplus_{pe}, \otimes_{pe})$ further forms a gyrovector space.*

## 4 MERITS OF CORRELATION MATRICES

The full-rank correlation matrices have been successfully applied across various fields (Varoquaux et al., 2010; Epskamp & Fried, 2018; Garba et al., 2021). However, their geometry has received far less attention compared to that of SPD matrices. A correlation matrix is obtained from the covariance matrix by dividing by the standard deviation of each variable. Specifically, for an invertible covariance matrix $P = (\mathrm{Cov}(X_i, X_j))_{1 \le i,j \le n} \in \mathrm{Sym}_n^+$ of a random vector $X$. the corresponding correlation matrix C is given by

$$\begin{aligned} C = \mathrm{Cor}(X_i, X_j) &= \frac{\mathrm{Cov}(X_i, X_j)}{\sqrt{\mathrm{Cov}(X_i, X_i)}\sqrt{\mathrm{Cov}(X_j, X_j)}} \\ &= \frac{P_{ij}}{\sqrt{P_{ii}}\sqrt{P_{jj}}} = [\mathrm{Diag}(P)^{-\frac{1}{2}} P\mathrm{Diag}(P)^{-\frac{1}{2}}]_{i,j}, \end{aligned} \tag{10}$$

where $\mathrm{Diag}(P)$ is the diagonal matrix of the same size as matrix P. Thus, it is often viewed as a quotient manifold of the SPD manifolds by the space of positive diagonal matrices and often considered as covariance matrices on which one can use the classical tools. It is unreasonable to directly apply the SPD metric to the full-rank correlation matrix. First, because correlation matrices have strong diagonal constraints (diagonal elements equal to one), SPD metrics cannot enforce these constraints. Additionally, correlation matrices are not stable under the action of the orthogonal group, unlike covariance matrices, which are invariant under transformations of the form $P \to O^T P O$, where $O$ is an orthogonal matrix. Consequently, O(n)-invariant metrics on SPD matrices, such as the AI metric, are not applicable to full-rank correlation matrices. Furthermore, compared to the SPD matrix space, the space of full-rank correlation matrices is more compact (Thanwerdas,

2024). For two different covariance matrices $P_1, P_2 \in \mathrm{Sym}_n^+$, using Eq. (10), their corresponding correlation matrix may be the same. For example, we choose

$$P = \begin{bmatrix} 4 & 2 \\ 2 & 1 \end{bmatrix} \quad \text{and} \quad Q = \begin{bmatrix} 1 & 0.5 \\ 0.5 & 0.5 \end{bmatrix}. \tag{11}$$

Their corresponding correlation matrices can be calculated as follows:

$$C_1 = \mathrm{Diag}\,(P)^{-\frac{1}{2}}\,P\mathrm{Diag}\,(P)^{-\frac{1}{2}} = \begin{bmatrix} 0.5 & 0 \\ 0 & 1 \end{bmatrix} \begin{bmatrix} 4 & 2 \\ 2 & 1 \end{bmatrix} \begin{bmatrix} 0.5 & 0 \\ 0 & 1 \end{bmatrix} = \begin{bmatrix} 1 & 1 \\ 1 & 1 \end{bmatrix}, \tag{12}$$

$$C_2 = \mathrm{Diag}\,(Q)^{-\frac{1}{2}}\,Q\mathrm{Diag}\,(Q)^{-\frac{1}{2}} = \begin{bmatrix} 1 & 0 \\ 0 & \sqrt{2} \end{bmatrix} \begin{bmatrix} 1 & 0.5 \\ 0.5 & 0.5 \end{bmatrix} \begin{bmatrix} 1 & 0 \\ 0 & \sqrt{2} \end{bmatrix} = \begin{bmatrix} 1 & 1 \\ 1 & 1 \end{bmatrix}. \tag{13}$$

Additionally, assume two pairs of points $P_1 \sim P_2$ and $Q_1 \sim Q_1$ on the SPD manifolds. Their corresponding correlation matrix will be the same, i.e.,

$$\mathrm{Diag}\,(P_1)^{-\frac{1}{2}}\,P_1\mathrm{Diag}\,(P_1)^{-\frac{1}{2}} = \mathrm{Diag}\,(P_2)^{-\frac{1}{2}}\,P_2\mathrm{Diag}\,(P_2)^{-\frac{1}{2}}, \tag{14}$$

$$\mathrm{Diag}\,(Q_1)^{-\frac{1}{2}}\,Q_1\mathrm{Diag}\,(Q_1)^{-\frac{1}{2}} = \mathrm{Diag}\,(Q_2)^{-\frac{1}{2}}\,Q_2\mathrm{Diag}\,(Q_2)^{-\frac{1}{2}}. \tag{15}$$

When using metrics on SPD manifolds, such as the widely used LE and LC metrics, the distance $d(P_1, Q_1) \neq d(P_2, Q_2)$. While the distances might be equal under the AI metric, they can differ when applying other operations. Let's take the logarithmic mapping as an example. Due to $T_{P_1}\mathrm{Sym}_n^+ \neq T_{P_1}\mathrm{Sym}_n^+$, it is straightforward to show that $Log_{P_1}Q_1 \neq Log_{P_2}Q_2$. Therefore, we use the metrics on SPD manifolds to operate it, and the results will be different. However, within the space of correlation matrices, $P1$ and $P2$ as well as $Q1$ and $Q2$ become the same correlation matrices. This implies that fundamental operations—such as matrix operations, exponential mappings, and logarithmic mappings—will yield the same results. This also demonstrates that the correlation matrix space is more compact, indicating that correlation matrices might capture more compact statistical information. This motivates us to explore correlation matrices and extend the gyrovector spaces to correlation matrix manifolds.

## 5 GYROVECTOR SPACES OF FULL-RANK CORRELATION MATRICES

Compared to SPD matrices, the geometry of full-rank correlation matrices has been less studied. In this subsection, we explore the gyro-structure of full-rank correlation matrix manifolds under Euclidean-Cholesky (EC) and log-Euclidean-Cholesky (LEC) geometries.

### 5.1 EC GYROVECTOR SPACES

The Cholesky map has already been applied to SPD matrices (Wang et al., 2003). To extend this to correlation matrices, the EC metric on full-rank correlation matrices was recently proposed (Thanwerdas & Pennec, 2022a) with much faster computations (Thanwerdas & Pennec, 2022a). Therefore, it is valuable to explore the gyro-structure of full-rank correlation matrix manifolds based on the EC geometry. Let $\mathrm{Diag}(P)$ denote the diagonal matrix of the same size as matrix P, and $\mathrm{Chol}(P)$ represents the lower triangular matrix obtained from the Cholesky decomposition of matrix $P \in \mathrm{Cor}_n^+$.

**Lemma 5.1.** [↓] *For $P, Q \in \mathrm{Cor}_n^+$, the binary operation $P \oplus_{ec} Q$ is given as*

$$P \oplus_{ec} Q = \Phi\left(\mathrm{Diag}\,(\mathrm{Chol}\,(P))^{-1}\,\mathrm{Chol}\,(P) + \mathrm{Diag}\,(\mathrm{Chol}\,(Q))^{-1}\,\mathrm{Chol}\,(Q) - I_n\right), \tag{16}$$

*where $\Phi(X) = \mathrm{Diag}(XX^T)^{-\frac{1}{2}}XX^T\mathrm{Diag}(XX^T)^{-\frac{1}{2}}$, $I_n$ is the $n \times n$ identity matrix.*

$I_n$ is the identity element of $\mathrm{Cor}_n^+$. Thus, using Eq. (16), we can obtain the inverse of P which is given by

$$\ominus_{ec}P = \Phi\left(2I_n - \mathrm{Diag}\,(\mathrm{Chol}\,(P))^{-1}\,\mathrm{Chol}\,(P)\right). \tag{17}$$

**Lemma 5.2.** [↓] *For $P \in \mathrm{Cor}_n^+$ and $t \in \mathbb{R}$, the scalar multiplication $t \otimes_{ec} P$ is given as*

$$t \otimes_{ec} P = \Phi\left(t\mathrm{Diag}\,(\mathrm{Chol}\,(P))^{-1}\,\mathrm{Chol}\,(P) + (1-t)\,I_n\right). \tag{18}$$

**Theorem 5.3.** [↓] *$(\mathrm{Cor}_n^+, \oplus_{ec})$ forms a gyrogroup. Endowed with the scalar multiplication $\otimes_{ec}$, $(\mathrm{Cor}_n^+, \oplus_{ec}, \otimes_{ec})$ further forms a gyrovector space.*

## 5.2 LEC Gyrovector Spaces

Since the matrix logarithm is a smooth diffeomorphism from $\mathrm{LT}_n^1$ to $\mathrm{LT}_n^0$, the EC metric can be extended to the Log-Euclidean-Cholesky (LEC) metric by matrix logarithm. Under this metric, we can obtain the associated gyro-structures.

**Lemma 5.4.** [↓] *For $P, Q \in \mathrm{Cor}_n^+$, the binary operation $P \oplus_{lec} Q$ is given as*

$$P \oplus_{lec} Q = \Phi \circ \exp\left(\log\left(\mathrm{Diag}\left(\mathrm{Chol}\left(P\right)\right)^{-1}\mathrm{Chol}\left(P\right)\right) + \log\left(\mathrm{Diag}\left(\mathrm{Chol}\left(Q\right)\right)^{-1}\mathrm{Chol}\left(Q\right)\right)\right), \quad (19)$$

*where $\Phi(X) = \mathrm{Diag}(XX^T)^{-\frac{1}{2}} XX^T \mathrm{Diag}(XX^T)^{-\frac{1}{2}}$, $I_n$ is the $n \times n$ identity matrix.*

$I_n$ is the identity element of $\mathrm{Cor}_n^+$. Thus, using Eq. (19), we can obtain the inverse of P which is given by

$$\ominus_{lec} P = P^{-1}. \quad (20)$$

**Lemma 5.5.** [↓] *For $P \in \mathrm{Cor}_n^+$ and $t \in \mathbb{R}$, the scalar multiplication $t \otimes_{lec} P$ is given as*

$$t \otimes_{lec} P = P^t. \quad (21)$$

**Theorem 5.6.** [↓] $(\mathrm{Cor}_n^+, \oplus_{lec})$ *forms a gyrogroup. Endowed with the scalar multiplication $\otimes_{lec}$, $(\mathrm{Cor}_n^+, \oplus_{lec}, \otimes_{lec})$ further forms a gyrovector space.*

## 6 Experiments

To demonstrate the effectiveness of the proposed method, we apply it to learning entity and relation embeddings within the SPD and full-rank correlation matrix manifolds for knowledge graph completion tasks.

### 6.1 Gyro for knowledge graph completion

**Problem Formulation:** Knowledge graphs (KGs) represent heterogeneous knowledge as triples of the form (subject, relation, object), where the subject and object denote entities, and the relation describes the interaction between them (Balazevic et al., 2019). KGs exhibit complex and diverse structures, with entities connected by symmetric, antisymmetric, or hierarchical relations. As KG is often incomplete, the goal is to predict missing links and identify valid but unobserved connections. Let $\mathcal{F} = (\mathcal{E}, \mathcal{R}, \mathcal{T})$ represent a knowledge graph, where $\mathcal{E}$ is the set of entities, $\mathcal{R}$ is the set of relations, and $\mathcal{T} \subseteq \mathcal{E} \times \mathcal{R} \times \mathcal{E}$ is the set of triples stored in the graph. The typical approach is to learn a scoring function $\phi : \mathcal{E} \times \mathcal{R} \times \mathcal{E} \to \mathbb{R}$ that evaluates the likelihood of a triple being true, enabling accurate ranking of missing triples. To achieve this, we propose learning entity embeddings within the SPD and full-rank correlation manifolds.

**Scoring Model:** Our model learns a scoring function given below

$$\phi(e_s, r, e_o) = -d\left((A \otimes S) \oplus R, O\right)^2 + b_s + b_o, \quad (22)$$

where $S, O$ represent embeddings, $b_s, b_o \in \mathbb{R}$ are respectively scalar biases for the subject and object entities, and $A, R$ are two matrices that depend on the relation $r$. The scaling transformation (matrix scaling) $\otimes$ is defined as

$$A \otimes S = \mathrm{Exp}_I(A * \mathrm{Log}_I(S)), \quad (23)$$

where $*$ denotes the Hadamard product, $A$ here signifies the tangent vector of matrix manifolds. For the SPD manifold, $A \in \mathrm{Sym}_n$, while $A \in \mathrm{Hol}_n$ in the case of full-rank correlation matrix manifold.

**Product manifold-based Fusion mechanism:** Our methods combine with the recently proposed gyro-structure of Grassmann manifolds (Nguyen, 2022). Inspired by Balazevic et al. (2019); Nguyen (2022), we train the entity embeddings on two product manifolds, *i.e.*, $\mathrm{Gr}_{n1,p} \times \mathrm{Sym}_{n2}^+$ and $\mathrm{Gr}_{n1,p} \times \mathrm{Cor}_{n2}^+$. The binary operation $\oplus$ on these manifolds is defined as

$$(P_{spd}, P_{gr}) \oplus (R_{spd}, R_{gr}) = (P_{spd} \oplus_{spd} R_{spd}, P_{gr} \oplus_{gr} R_{gr}), \quad (24)$$

$$(P_{cor}, P_{gr}) \oplus (R_{cor}, R_{gr}) = (P_{cor} \oplus_{cor} R_{cor}, P_{gr} \oplus_{gr} R_{gr}), \quad (25)$$

where $P_{spd}, R_{spd} \in \mathrm{Sym}_{n2}^+$, $P_{cor}, R_{cor} \in \mathrm{Cor}_{n2}^+$, and $P_{gr}, R_{gr} \in \mathrm{Gr}_{n1,p}^+$. The symbols $\oplus_{spd}$ and $\oplus_{cor}$ respectively denote the binary operations under the SPD and full-rank correlation matrix manifolds, while $\oplus_{gr}$ signifies that of the Grassmann manifolds (Nguyen, 2022, Sec 3.2).

Table 1: The necessary operators in the network based on the different metrics.

| Manifold | SPD | full-rank correlation matrices | |
|---|---|---|---|
| Metric | PE metric | EC metric | LEC metric |
| Tangent vector | $\mathrm{Sym}_n$ | $\mathrm{Hol}_n$ | $\mathrm{Hol}_n$ |
| $\mathrm{Exp}_I(W)$ | $(P^\alpha + I_n)^{\frac{1}{\alpha}}$ | $\Phi\left(I_n + \lfloor W \rfloor\right)$ | $\exp(W)$ |
| $P \oplus R$ | $(P^\alpha + R^\alpha - I_n)^{\frac{1}{\alpha}}$ | $\Phi\left(\Theta(P) + \Theta(R) - I_n\right)$ | $\Phi \circ \exp\left(\log\left(\Theta(P)\right) + \log\left(\Theta(R)\right)\right)$ |
| $A \otimes S$ | $(AS^\alpha + I_n - A)^{\frac{1}{\alpha}}$ | $\Phi\left(A\Theta(S) + I_n - A\right)$ | $\Phi \circ \exp\left(A\log\left(\Theta(S)\right)\right)$ |
| $d(P,R)$ | $\frac{1}{\alpha}\|P^\alpha - R^\alpha\|$ | $\|\Theta(R) - \Theta(P)\|$ | $\|\log\left(\Theta(R)\right) - \log\left(\Theta(P)\right)\|$ |

The corresponding scalar operation $\otimes$ on these manifolds is formulated as

$$(A_{spd}, A_{gr}) \otimes (S_{spd}, S_{gr}) = (A_{spd} \otimes_{spd} S_{spd}, A_{gr} \otimes_{gr} S_{gr}), \tag{26}$$

$$(A_{cor}, A_{gr}) \otimes (S_{cor}, S_{gr}) = (A_{cor} \otimes_{cor} S_{cor}, A_{gr} \otimes_{gr} S_{gr}), \tag{27}$$

where $S_{spd} \in \mathrm{Sym}_{n2}^+$, $S_{cor} \in \mathrm{Cor}_{n2}^+$, $S_{gr} \in \mathrm{Gr}_{n1,p}^+$, $A_{spd} \in \mathrm{Sym}_{n2}$, $A_{cor} \in \mathrm{Hol}_{n2}$, and $A_{gr} \in \mathrm{Gr}_{p,n1-p}^+$ are six matrices associated with the relation $r$. The symbols $\otimes_{spd}$ and $\otimes_{cor}$ respectively represent matrix scaling operations under the SPD and full-rank correlation matrix manifolds, while $\otimes_{gr}$ denotes the corresponding operation on the Grassmann manifolds (Nguyen, 2022, Sec. 4.2.2).

Similarly, the distance functions $d(\cdot, \cdot)$ on these manifolds are defined as

$$d\left((P_{spd}, P_{gr}), (R_{spd}, R_{gr})\right) = \eta d_{spd}\left(P_{spd}, R_{spd}\right) + d_{gr}\left(P_{gr}, R_{gr}\right), \tag{28}$$

$$d\left((P_{cor}, P_{gr}), (R_{cor}, R_{gr})\right) = \eta d_{cor}\left(P_{cor}, R_{cor}\right) + d_{gr}\left(P_{gr}, R_{gr}\right), \tag{29}$$

where $\eta$ is a constant, $d_{spd}(\cdot, \cdot)$ and $d_{cor}(\cdot, \cdot)$ respectively signify the distance functions under the SPD and full-rank correlation matrix manifolds, while $d_{gr}(\cdot, \cdot)$ represents the corresponding function under the Grassmann manifolds (Nguyen, 2022, Sec 4.2.2). We summarize the necessary operators involved in the proposed model in Tab. 1, where $\Theta(X) = \mathrm{Diag}\left(\mathrm{Chol}(X)\right)^{-1}\mathrm{Chol}(X)$ and $\lfloor W \rfloor$ denotes the strictly lower triangular terms of $W$.

## 6.2 Datasets and experimental settings

We evaluate the performance of the proposed method on two benchmarking datasets: WN18RR (Bordes et al., 2013; Dettmers et al., 2018) and FB15k-237(Bordes et al., 2013; Toutanova & Chen, 2015). The WN18RR dataset is a subset of WordNet (Miller, 1995), a hierarchical collection of relations between words. It was created from WN18 (Bordes et al., 2013) by removing the inverse of many relations from the validation and test sets, making the dataset more challenging. This datasets consists of 93,003 triples, featuring 40,943 entities and 11 relations. The FB15k-237 dataset, a subset of Freebase (Bollacker et al., 2008), is derived in a manner akin to WN18RR (Bordes et al., 2013), containing 14,541 entities and 237 relations.

Following the criterion in López et al. (2021), we use binary cross-entropy loss as the training objective and AdamW optimizer for 5000 epochs. Each batch contains 4,096 samples, with 10 negative samples per positive instance. For evaluation, we adopt the mean reciprocal rank (MRR) and hits at K (H@K, where K = 1, 3, 10) metrics to assess the proportion of correctly ranked entities within the top K positions (López et al., 2021; Nguyen, 2022). To ensure model efficiency, early stopping is triggered when the MRR on the validation set does not improve after 500 consecutive epochs. The model checkpoint with the highest MRR on the validation set are used for subsequent testing. For our proposed product manifold-based models, we set the candidate sets of $\eta$, learning rate, and weight decay to $\{0.5, 1, 1.5\}$, $\{5e-4, 1e-3, 2.5e-3\}$, and $\{1e-2, 1e-3, 1e-4, 1e-5\}$, respectively. All experiments are run on a PC equipped with an i9-13900HX CPU and 16GB of RAM. Despite the code's compatibility with GPUs, leveraging GPU resources does not expedite training due to the dominance of eigenvalue operations, which is identified as the primary computational bottleneck (López et al., 2021).

Drawing inspiration from Chami et al. (2019); López et al. (2021), we use trivialization via the Riemannian exponential map to optimize non-Euclidean parameters (Lezcano Casado, 2019). Specifically, each SPD parameter is modeled by a symmetric matrix via the exponential map at the identity

Table 2: Results on the WN18RR dataset

| Model | MRR | H@1 | H@3 | H@10 | Time (seconds) | |
|---|---|---|---|---|---|---|
| | | | | | Train/epoch | Test |
| $\text{SPD}_{Sca}^{R}$ | 41.7 | 36.5 | 44.5 | 51.1 | 29.6 | 621.0 |
| $\text{SPD}_{Sca}^{F_1}$ | 40.8 | 36.3 | 42.9 | 49.5 | 29.6 | 617.2 |
| $\text{SPD}_{Rot}^{R}$ | 22.4 | 8.4 | 33.4 | 47.3 | 29.8 | 618.9 |
| $\text{SPD}_{Rot}^{F_1}$ | 26.5 | 18.1 | 30.7 | 42.9 | 29.8 | 616.5 |
| $\text{SPD}_{Ref}^{R}$ | 41.0 | 37.1 | 42.7 | 47.6 | 30.1 | 613.1 |
| $\text{SPD}_{Ref}^{F_1}$ | 39.7 | 35.9 | 41.5 | 46.3 | 30.1 | 611.4 |
| GyroGRLE-KGCNet | 41.5 | 35.3 | 44.9 | 52.1 | 9.5 | **5.4** |
| GyroECGR-KGCNet | 42.6 | 37.3 | 45.2 | 51.6 | **9.0** | 9.2 |
| GyroLECGR-KGCNet | 42.9 | 37.6 | 45.9 | 51.8 | 10.0 | 14,6 |
| GyroPEGR-KGCNet | **44.4** | **39.3** | **46.8** | **53.4** | 27.2 | 14.7 |

Table 3: Results on the FB15k-237 dataset

| Model | MRR | H@1 | H@3 | H@10 | Time (seconds) | |
|---|---|---|---|---|---|---|
| | | | | | Train/epoch | Test |
| $\text{SPD}_{Sca}^{R}$ | 29.5 | 21.0 | 32.3 | 46.8 | 91.9 | 1435.2 |
| $\text{SPD}_{Sca}^{F_1}$ | 29.1 | 19.7 | 30.6 | 45.1 | 91.9 | 1445.9 |
| $\text{SPD}_{Rot}^{R}$ | 29.0 | 20.2 | 31.7 | 46.5 | 92.7 | 1419.4 |
| $\text{SPD}_{Rot}^{F_1}$ | 27.8 | 19.0 | 30.3 | 44.7 | 92.7 | 1416.8 |
| $\text{SPD}_{Ref}^{R}$ | 28.1 | 19.9 | 30.6 | 44.7 | 92.9 | 1410.8 |
| $\text{SPD}_{Ref}^{F_1}$ | 27.1 | 19.0 | 29.4 | 43.1 | 92.9 | 1408.6 |
| GyroGRLE-KGCNet | 29.2 | 20.5 | 32.1 | 46.7 | **26.8** | **12.3** |
| GyroECGR-KGCNet | **29.8** | **21.2** | 32.5 | 47.0 | 27.4 | 21.4 |
| GyroLECGR-KGCNet | 29.6 | 21.0 | 32.4 | 47.0 | 27.6 | 27.8 |
| GyroPEGR-KGCNet | 29.7 | 21.0 | **32.6** | **47.3** | 82.4 | 32.5 |

matrix. For full-rank correlation parameters, we model them in the tangent space at the identity, *i.e.,* $T_I \text{Cor}_n^+ \cong \text{Hol}_n$, followed by applying the exponential map at this point. This formulation enables all parameters to be optimized through standard Euclidean techniques, thereby circumventing the numerical instability often associated with Riemannian optimization (Bécigneul & Ganea, 2018; López et al., 2021).

## 6.3 RESULT

We compare our method with SPD models, which use three feature transformations (scaling, rotation or reflection) and two distance metrics (the Finsler One metric and the Riemannian metric) (López et al., 2021), as well as with GyroGRLE-KGCNet (Nguyen, 2022). Following Nguyen (2022), our models with 21 degrees of freedom (DOF) learn embeddings in $\text{Sym}_5^+ \times \text{Gr}_{5,2}$ and $\text{Cor}_5^+ \times \text{Gr}_{5,2}$. For fairness, we keep the DOF of all models equal, allowing SPD models to learn embeddings in $\text{Sym}_6^+$ and GyroGRLE-KGCNet to learn embeddings in $\text{Sym}_5^+ \times \text{Gr}_{5,2}$. Tab. 2 and Tab. 3 represent the result of our models on the WN18RR and FB15k-237 datasets. Our models outperform the SPD models and GyroGRLE-KGCNet in the WN18RR and FB15k-237 datasets, achieving higher MRR, H@1, H@3, and H@10 scores. The proposed model GyroPEGR-KGCNet shows a significant improvement over GyroGRLE-KGCNet on both datasets, with MRR, H@1, H@3, and H@10 being 2.9%, 4.0%, 1.9%, and 1.3% higher on the WN18RR dataset, and 0.5%, 0.5%, 0.5%, and 0.6% higher on the FB15k-237 dataset. Compared to the SPD models, our models have a clear advantage in computation time. However, our GyroPEGR-KGCNet model requires more time than the GyroGRLE-KGCNet model due to the involvement of Singular Value Decomposition (SVD). Specifically, when learning the score function in each epoch, it involves four SVD-based matrix functions with the time complexity of $O(4dn^3)$, where $d$, $n$ denote batch size, the matrix dimension, respectively. Additionally, the proposed model based on the correlation matrix achieves higher scores than the SPD models and GyroGRLE-KGCNet, which are modeled on SPD spaces, in most

cases. This further demonstrates that correlation matrix spaces can capture richer geometric information.

### 6.4 ABLATIONS

Table 4: Results on the WN18RR dataset

| Model | MRR | H@1 | H@3 | H@10 | Time (seconds) | |
|---|---|---|---|---|---|---|
| | | | | | Train/epoch | Test |
| GyroLE-KGCNet | 37.8 | 33.4 | 39.9 | 45.2 | **2.0** | **3.1** |
| GyroGR-KGCNet | 11.5 | 5.9 | 11.1 | 25.0 | 6.5 | 2.8 |
| GyroEC-KGCNet | 31.9 | 23.5 | 37.2 | 46.4 | 3.3 | 7.4 |
| GyroLEC-KGCNet | 31.8 | 23.2 | 37.0 | 47.3 | 3.4 | 8.8 |
| GyroPE-KGCNet | **39.6** | **34.0** | **43.0** | **48.8** | 20.8 | 12.0 |

Table 5: Results on the FB15k-237 dataset

| Model | MRR | H@1 | H@3 | H@10 | Time (seconds) | |
|---|---|---|---|---|---|---|
| | | | | | Train/epoch | Test |
| GyroLE-KGCNet | 26.0 | 17.7 | 28.3 | 43.1 | **7.0** | **5.4** |
| GyroGR-KGCNet | 18.3 | 12.6 | 19.6 | 30.0 | 20.8 | 7.6 |
| GyroEC-KGCNet | 27.0 | 18.8 | 29.5 | 43.5 | 7.8 | 14.7 |
| GyroLEC-KGCNet | 26.9 | 18.7 | 29.5 | 43.3 | 8.0 | 19.4 |
| GyroPE-KGCNet | **28.7** | **20.2** | **31.3** | **45.4** | 62.4 | 26.8 |

To further illustrate the effectiveness of our method, we implemented models that learn embeddings in $\mathrm{Sym}_5^+$ and $\mathrm{Cor}_5^+$. A notable observation from Tab. 4 and Tab. 5 is that, compared to GyroLE-KGCNet and GyroGR-KGCNet, which learn embeddings in $\mathrm{Sym}_5^+$ and $\mathrm{Gr}_{5,2}$, respectively, our models demonstrate superior results on the WN18RR and FB15k-237 datasets. The performance improvements are significant in all cases, highlighting the effectiveness of our approach.

Table 6: Results on the WN18RR dataset

| Model | MRR | H@1 | H@3 | H@10 | Time (seconds) | |
|---|---|---|---|---|---|---|
| | | | | | Train/epoch | Test |
| GyroECGR-KGCNet | 42.6 | 37.3 | 45.2 | 51.6 | **9.0** | **9.2** |
| GyroLECGR-KGCNet | 42.9 | 37.6 | 45.9 | 51.8 | 10.0 | 14.6 |
| GyroPEGR-KGCNet | 44.4 | 39.3 | 46.8 | 53.4 | 27.2 | 14.7 |
| GyroPRECGR-KGCNet | **45.3** | 39.7 | **47.8** | **54.9** | 30.9 | 25.2 |
| GyroPELECGR-KGCNet | 45.2 | **39.8** | 47.7 | 54.8 | 31.0 | 27.4 |

In addition to learning embeddings in $\mathrm{Sym}_5^+ \times \mathrm{Gr}_{5,2}$ or $\mathrm{Cor}_5^+ \times \mathrm{Gr}_{5,2}$, we also explore learning embeddings in $\mathrm{Sym}_5^+ \times \mathrm{Sym}_5^+ \times \mathrm{Gr}_{5,2}$. We conduct experiments on the WN81RR dataset, and the results are shown in Tab. 6. We can observe that the models GyroPEECGR-KGCNet and GyroPELECGR-KGCNet which learn learning embeddings in $\mathrm{Sym}_5^+ \times \mathrm{Sym}_5^+ \times \mathrm{Gr}_{5,2}$, improve the proposed models that learn embeddings in $\mathrm{Sym}_5^+ \times \mathrm{Gr}_{5,2}$ or $\mathrm{Cor}_5^+ \times \mathrm{Gr}_{5,2}$ in all the cases. This proves the effectiveness of embeddings in product spaces of SPD, full-rank correlation matrices and Grassmann manifolds.

## 7 CONCLUSION

In this paper, we extend the SPD gyro-structure into the PE geometry which recovers the existing LE gyro space when power tends to 0. Besides, we also explore the geometric structure of correlation matrices and propose two novel gyro-structures for full-rank correlation matrices, induced by the theoretically and computationally convenient EC and LEC metrics. Extensive experiments on knowledge graph completion tasks demonstrate the effectiveness of our algebraic structures.

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

## A NOTIONS

For better clarity, we summarize all the notations used in this paper in Tab. 7.

Table 7: Summary of notations.

| Symbol | Explanation |
|---|---|
| $\mathrm{M}_{n,m}$ | Space of $n \times m$ matrices |
| $\mathrm{Sym}_n^+$ | Space of $n \times n$ SPD matrices |
| $\mathrm{Sym}_n$ | Space of $n \times n$ symmetric matrices |
| $\mathrm{Sym}_n^{+,pe}$ | Space of $n \times n$ SPD matrices with PE geometry |
| $\mathrm{Cor}_n^+$ | Space of $n \times n$ full-rank correlation matrices |
| $\mathrm{Cor}_n^{+,ec}$ | Space of $n \times n$ full-rank correlation matrices with EC geometry |
| $\mathrm{Cor}_n^{+,lec}$ | Space of $n \times n$ full-rank correlation matrices with LEC geometry |
| $\mathrm{Hol}_n$ | Space of $n \times n$ symmetric matrices which diagonal elements are 0 |
| $\mathrm{LT}_n$ | Space of the lower triangular part of $n \times n$ matrices |
| $\mathrm{LT}_n^0$ | $\mathrm{LT}_n$ with null diagonal elements |
| $\mathrm{LT}_n^1$ | $\mathrm{LT}_n$ with unit diagonal elements |
| $\mathcal{M}$ | Homogeneous Riemannian manifold |
| $T_P\mathcal{M}$ | Tangent space at $P \in \mathcal{M}$ |
| $\exp(P)$ | Matrix exponential of $P$ |
| $\log(P)$ | Matrix logarithm of $P$ |
| $\mathrm{Chol}(P)$ | Cholesky decomposition of matrix of $P$ |
| $\lfloor P \rfloor$ | the strictly lower triangular terms of $P$ |
| $\mathrm{Diag}(P)$ | Diagonal part of matrix of $P$ |
| $\mathrm{Exp}_P(\cdot)$ | Riemannian exponential map at $P \in \mathcal{M}$ |
| $\mathrm{Log}_P(\cdot)$ | Riemannian logarithmic map at $P \in \mathcal{M}$ |
| $\Gamma_{P \to Q}(W)$ | Parallel transport of $W$ from $P$ to $Q$ in $\mathcal{M}$ |
| $d(\cdot, \cdot)$ | Geodesic distance |
| $g_P(\cdot, \cdot)$ | Riemannian metric at $P \in \mathcal{M}$ |
| $\oplus_{pe}, \ominus_{pe}$ | Binary and inverse operations in $\mathrm{Sym}_n^{+,pe}$ |
| $\otimes_{pe}$ | Scalar multiplication operations in $\mathrm{Sym}_n^{+,pe}$ |
| $\oplus_{ec}, \ominus_{ec}$ | Binary and inverse operations in $\mathrm{Cor}_n^{+,ec}$ |
| $\otimes_{ec}$ | Scalar multiplication operations in $\mathrm{Cor}_n^{+,ec}$ |
| $\oplus_{lec}, \ominus_{lec}$ | Binary and inverse operations in $\mathrm{Cor}_n^{+,lec}$ |
| $\otimes_{lec}$ | Scalar multiplication operations in $\mathrm{Cor}_n^{+,lec}$ |

## B THE BASIC OPERATIONS BASED ON THE DIFFERENT METRICS

**PE metric**: For $P, Q \in \mathrm{Sym}_n^+, W \in T_P\mathrm{Sym}_n^+$, the key operations based on the PE metric (Dryden et al., 2010) is given by

$$\text{Riemannian distance: } d(P, Q) = \frac{1}{\alpha}\|P^\alpha - Q^\alpha\|, \tag{30}$$

$$\text{Riemannian exponential map: } \mathrm{Exp}_P(W) = (P^\alpha + d_P\mathrm{pow}_\alpha(W))^{\frac{1}{\alpha}}, \tag{31}$$

$$\text{Riemannian logarithmic map: } \mathrm{Log}_P(Q) = (d_P\mathrm{pow}_\alpha)^{-1}(Q^\alpha - P^\alpha), \tag{32}$$

$$\text{Parallel transport: } \Gamma_{P \to Q}(W) = (d_Q\mathrm{pow}_\alpha)^{-1}(d_P\mathrm{pow}_\alpha(W)). \tag{33}$$

**EC metric**: For $P, Q \in \mathrm{Cor}_n^+, W \in T_P\mathrm{Cor}_n^+$, the key operations based on the EC metric (Thanwerdas & Pennec, 2022b) is given by

$$\text{Riemannian distance: } d(P, Q) = \|\Theta(Q) - \Theta(P)\|, \tag{34}$$

$$\text{Riemannian exponential map: } \mathrm{Exp}_P(W) = \Theta^{-1}(\Theta(P) + d_P\Theta(W)), \tag{35}$$

$$\text{Riemannian logarithmic map: } \mathrm{Log}_P(Q) = (d_P\Theta)^{-1}(\Theta(Q) - \Theta(P)), \tag{36}$$

$$\text{Parallel transport: } \Gamma_{P \to Q}(W) = (d_Q\Theta)^{-1}(d_P\Theta(W)), \tag{37}$$

where $\Theta(P) = \mathrm{Diag}\left(\mathrm{Chol}\left(P\right)\right)^{-1}\mathrm{Chol}\left(P\right).$

**LEC metric**: For $P, Q \in \mathrm{Cor}_n^+, W \in T_P\mathrm{Cor}_n^+$, the key operations based on the LEC metric (Thanwerdas & Pennec, 2022b) is given by

Riemannian distance: $d(P, Q) = \|\log\left(\Theta(Q)\right) - \log\left(\Theta(P)\right)\|,$ $\qquad$ (38)

Riemannian exponential map: $\mathrm{Exp}_P(W) = \Theta^{-1} \circ \exp\left(\log\left(\Theta\left(P\right)\right) + d_P\left(\log \circ\Theta\right)(W)\right),$ $\quad$ (39)

Riemannian logarithmic map: $\mathrm{Log}_P(Q) = \left(d_P\left(\log \circ\Theta\right)\right)^{-1}\left(\log\left(\Theta\left(Q\right)\right) - \log\left(\Theta\left(P\right)\right)\right),$ $\quad$ (40)

Parallel transport: $\Gamma_{P\to Q}(W) = \left(d_Q\left(\log \circ\Theta\right)\right)^{-1}\left(d_P\log \circ\Theta\left(W\right)\right),$ $\qquad$ (41)

where $\Theta(P) = \mathrm{Diag}\left(\mathrm{Chol}\left(P\right)\right)^{-1}\mathrm{Chol}\left(P\right).$

## C   PROOFS OF THE PROPOSITIONS AND THEORIES IN THE MAIN PAPER

*Proof of Lem. 3.1* . Using the basic operations based on the PE metric , we can deduce that

$$P \oplus_{pe} Q = \mathrm{Exp}_P\left(\Gamma_{I_n \to P}\left(\mathrm{Log}_{I_n}\left(Q\right)\right)\right)$$
$$= \mathrm{Exp}_P\left(\left(d_p\mathrm{pow}_\alpha\right)^{-1}\left(Q^\alpha - I_n\right)\right) \qquad (42)$$
$$= \left(P^\alpha + Q^\alpha - I_n\right)^{\frac{1}{\alpha}}.$$

$\square$

*Proof of Lem. 3.2* . Using Eqs. (31) and (32), it is straightforward to see that

$$t \otimes P = \mathrm{Exp}_{I_n}\left(t\mathrm{Log}_{I_n}\left(P\right)\right)$$
$$= \mathrm{Exp}_{I_n}\left(\left(d_{I_n}\mathrm{pow}_\alpha\right)^{-1}\left(t\left(P^\alpha - I_n\right)\right)\right) \qquad (43)$$
$$= \left(tP^\alpha + (1-t)I_n\right)^{\frac{1}{\alpha}}.$$

$\square$

*Proof of Thm. 3.3* . The gyroautomorphism can be determined from the binary operations (Ungar, 2005; 2012; 2014). Using Eq. (6), we can deduce that

$$\mathrm{gyr}_{pe}[P, Q]R = \left(\ominus_{pe}\left(P \oplus_{pe} Q\right)\right) \oplus_{pe} \left(P \oplus_{pe} \left(Q \oplus_{pe} R\right)\right)$$
$$\overset{(1)}{=} \left(\ominus_{pe}\left(P^\alpha + Q^\alpha - I_n\right)^{\frac{1}{\alpha}}\right) \oplus_{pe} \left(P \oplus_{pe}\left(Q^\alpha + R^\alpha - I_n\right)^{\frac{1}{\alpha}}\right)$$
$$\overset{(2)}{=} \left(3I_n - P^\alpha - Q^\alpha\right)^{\frac{1}{\alpha}} \oplus_{pe} \left(P^\alpha + Q^\alpha + R^\alpha - 2I_n\right)^{\frac{1}{\alpha}} \qquad (44)$$
$$\overset{(3)}{=} \left(3I_n - P^\alpha - Q^\alpha + P^\alpha + Q^\alpha + R^\alpha - 2I_n - I_n\right)^{\frac{1}{\alpha}}$$
$$= R.$$

The above derivation comes from the following.

(1) and (3) follow from Eq. (7).

(2) follows from Eq. (8).

We can deduce that $\mathrm{gyr}_{pe}[a, b] = I_d$. Next, we will prove that $(\mathrm{Sym}_n^+, \oplus_{pe})$ forms a gyrogroup, *i.e.*, it satisfies axioms G1, G2, G3, G4 and Gyrocommutative Law.

**Axiom (G1)**

*Proof.* For $P \in \mathrm{Sym}_n^+$, we have

$$I_n \oplus_{pe} P = \left(P^\alpha + I_n - I_n\right)^{\frac{1}{\alpha}} = P. \qquad (45)$$

Therefore, $I_n$ is a left identity in $\mathrm{Sym}_n^+$. $\qquad\square$

**Axiom (G2)**

*Proof.* For $P \in \mathrm{Sym}_n^+$, let $Q = (2I_n - P^\alpha)^{\frac{1}{\alpha}} \in \mathrm{Sym}_n^+$, we have

$$
\begin{aligned}
Q \oplus_{pe} P &= \left( P^\alpha + \left( (2I_n - P^\alpha)^{\frac{1}{\alpha}} \right)^\alpha - I_n \right)^{\frac{1}{\alpha}} \\
&= (P^\alpha + 2I_n - P^\alpha - I_n)^{\frac{1}{\alpha}} \\
&= I_n.
\end{aligned}
\tag{46}
$$

Therefore, $Q = (2I_n - P^\alpha)^{\frac{1}{\alpha}}$ is a left inverse of $P$. $\qquad\square$

**Axiom (G3)**

*Proof.* For $P, Q, R \in \mathrm{Sym}_n^+$, we have

$$
\begin{aligned}
P \oplus_{pe} (Q \oplus_{pe} R) &= P \oplus_{pe} (Q^\alpha + R^\alpha - I_n)^{\frac{1}{\alpha}} \\
&= (P^\alpha + Q^\alpha + R^\alpha - 2I_n)^{\frac{1}{\alpha}}.
\end{aligned}
\tag{47}
$$

Since $\mathrm{gyr}_{pe}[P, Q]R = R$, we have

$$
\begin{aligned}
(P \oplus_{pe} Q) \oplus_{pe} \mathrm{gyr}_{pe}[P, Q]R &= (P \oplus_{pe} Q) \oplus_{pe} R \\
&= (P^\alpha + Q^\alpha - I_n)^{\frac{1}{\alpha}} \oplus_{pe} R \\
&= (P^\alpha + Q^\alpha + R^\alpha - 2I_n)^{\frac{1}{\alpha}}.
\end{aligned}
\tag{48}
$$

Therefore, $P \oplus_{pe} (Q \oplus_{pe} R) = (P \oplus_{pe} Q) \oplus_{pe} \mathrm{gyr}_{pe}[P, Q]R$. $\qquad\square$

**Axiom (G4)**

*Proof.* For $P, Q, R \in \mathrm{Sym}_n^+$, we have

$$
\begin{aligned}
&\mathrm{gyr}_{pe}[P \oplus_{pe} Q, Q]R \\
&= (\ominus_{pe} (P \oplus_{pe} Q \oplus_{pe} Q)) \oplus_{pe} ((P \oplus_{pe} Q) \oplus_{pe} (Q \oplus_{pe} R)) \\
&= \left( \ominus_{pe} (P^\alpha + 2Q^\alpha - 2I_n)^{\frac{1}{\alpha}} \right) \oplus_{pe} \left( (P^\alpha + Q^\alpha - I_n)^{\frac{1}{\alpha}} \oplus_{pe} (Q^\alpha + R^\alpha - I_n)^{\frac{1}{\alpha}} \right) \\
&= (4I_n - P^\alpha - 2Q^\alpha)^{\frac{1}{\alpha}} \oplus_{pe} (P^\alpha + 2Q^\alpha + R^\alpha - 3I_n)^{\frac{1}{\alpha}} \\
&= (4I_n - P^\alpha - 2Q^\alpha + P^\alpha + 2Q^\alpha + R^\alpha - 3I_n - I_n)^{\frac{1}{\alpha}} \\
&= R.
\end{aligned}
\tag{49}
$$

Therefore, $\mathrm{gyr}_{pe}[P, Q]R = \mathrm{gyr}_{pe}[P \oplus_{pe} Q, Q]R$. $\qquad\square$

**Gyrocommutative Law**

*Proof.* Since we have proved that $\mathrm{gyr}_{pe}[P, Q] = I_d$, we have

$$
\begin{aligned}
\mathrm{gyr}_{pe}[P, Q](Q \oplus_{pe} P) &= Q \oplus_{pe} P \\
&= (Q^\alpha + P^\alpha - I_n)^{\frac{1}{\alpha}} \\
&= P \oplus_{pe} Q.
\end{aligned}
\tag{50}
$$

Therefore, it satisfies $P \oplus_{pe} Q = \mathrm{gyr}_{pe}[P, Q](Q \oplus_{pe} P)$. $\qquad\square$

Thus, $(\mathrm{Sym}_n^+, \oplus_{pe})$ forms a gyrogroup. Then, we will prove $(\mathrm{Sym}_n^+, \oplus_{pe}, \otimes_{pe})$ that endowed with the scalar multiplication, further forms a gyrovector space i.e., satisfying axioms V1, V2, V3, V4, V5 for gyrovector spaces.

**Axiom (V1)**

*Proof.* For $t \in \mathbb{R}$ and $P \in \mathrm{Sym}_n^+$, we have

$$1 \otimes_{pe} P = (P^\alpha + (1-1) I_n)^{\frac{1}{\alpha}} = P. \tag{51}$$

$$0 \otimes_{pe} P = (I_n)^{\frac{1}{\alpha}} = I_n. \tag{52}$$

$$t \otimes_{pe} I_n = (tI_n^\alpha + (1-t) I_n)^{\frac{1}{\alpha}} = I_n. \tag{53}$$

$$-1 \otimes_{pe} P = (-P^\alpha + (1-(-1)) I_n)^{\frac{1}{\alpha}} = \ominus_{pe} P. \tag{54}$$

$\square$

**Axiom (V2)**

*Proof.* For $s, t \in \mathbb{R}$ and $P \in \mathrm{Sym}_n^+$, we have

$$(s+t) \otimes_{pe} P = ((s+t) P^\alpha + (1 - (s+t)) I_n)^{\frac{1}{\alpha}} \tag{55}$$

$$(s \otimes_{pe} P) \oplus_{pe} (t \otimes_{pe} P) = (sP^\alpha + (1-s) I_n)^{\frac{1}{\alpha}} \oplus_{pe} (tP^\alpha + (1-t) I_n)^{\frac{1}{\alpha}}$$
$$= (sP^\alpha + (1-s) I_n + tP^\alpha + (1-t) I_n - I_n)^{\frac{1}{\alpha}} \tag{56}$$
$$= ((s+t) P^\alpha + (1 - (s+t)) I_n)^{\frac{1}{\alpha}}.$$

Therefore, $(s+t) \otimes_{pe} P = (s \otimes_{pe} P) \oplus_{pe} (t \otimes_{pe} P)$. $\square$

**Axiom (V3)**

*Proof.*

$$(st) \otimes_{pe} P = (stP^\alpha + (1-st) I_n)^{\frac{1}{\alpha}}. \tag{57}$$

$$s \otimes_{pe} (t \otimes_{pe} P) = s \otimes_{pe} (tP^\alpha + (1-t) I_n)^{\frac{1}{\alpha}}$$
$$= (s (tP^\alpha + (1-t) I_n) + (1-s) I_n)^{\frac{1}{\alpha}} \tag{58}$$
$$= (stP^\alpha + (1-st) I_n)^{\frac{1}{\alpha}}.$$

Therefore, $(st) \otimes_{pe} P = s \otimes_{pe} (t \otimes_{pe} P)$. $\square$

**Axiom (V4)**

*Proof.* For $t \in \mathbb{R}$ and $P, Q, R \in \mathrm{Sym}_n^+$, since we have proved that $\mathrm{gyr}_{pe}[P,Q] = I_d$, we have

$$\mathrm{gyr}_{pe}[P,Q] (t \otimes_{pe} R) = t \otimes_{pe} R = t \otimes_{pe} \mathrm{gyr}_{pe}[P,Q]R. \tag{59}$$

$\square$

**Axiom (V5)**

*Proof.* For $s, t \in \mathbb{R}$ and $P, R \in \mathrm{Sym}_n^+$, we have

$$\mathrm{gyr}_{pe}[s \otimes_{pe} P, t \otimes_{pe} P]R$$
$$= (\ominus_{pe} ((s \otimes_{pe} P) \oplus_{pe} (t \otimes_{pe} P))) \oplus_{pe} ((s \otimes_{pe} P) \oplus_{pe} (t \otimes_{pe} P \oplus_{pe} R))$$
$$= (\ominus_{pe} ((s+t) \otimes_{pe} P)) \oplus_{pe} \left( (sP^\alpha + (1-s) I_n)^{\frac{1}{\alpha}} \oplus_{pe} (tP^\alpha + (1-t) I_n + R^\alpha - I_n)^{\frac{1}{\alpha}} \right)$$
$$= (- (s+t) P^\alpha + (s+t+1) I_n)^{\frac{1}{\alpha}} \oplus_{pe} ((s+t) P^\alpha - (s+t) I_n + R^\alpha)^{\frac{1}{\alpha}} \tag{60}$$
$$= (- (s+t) P^\alpha + (s+t+1) I_n + (s+t) P^\alpha - (s+t) I_n + R^\alpha - I_n)^{\frac{1}{\alpha}}$$
$$= R.$$

Therefore, $\mathrm{gyr}_{pe}[s \otimes_{pe} P, t \otimes_{pe} P]R = I_d$ $\square$

Thus, $(\mathrm{Sym}_n^+, \oplus_{pe}, \otimes_{pe})$ further forms a gyrovector space. $\square$

*Proof of Lem. 5.1* . Using the basic operations based on the EC metric , we can deduce that

$$
\begin{aligned}
P \oplus_{ec} Q &= \mathrm{Exp}_P \left( \Gamma_{I_n \to P} \left( \mathrm{Log}_{I_n} (Q) \right) \right) \\
&= \mathrm{Exp}_P \left( (d_P \Theta)^{-1} \left( \Theta (Q) - I_n \right) \right). \\
&= \Theta^{-1} \left( \Theta (P) + \Theta (Q) - I_n \right) \\
&= \Phi \left( \mathrm{Diag} \left( \mathrm{Chol} (P) \right)^{-1} \mathrm{Chol} (P) + \mathrm{Diag} \left( \mathrm{Chol} (Q) \right)^{-1} \mathrm{Chol} (Q) - I_n \right),
\end{aligned}
\tag{61}
$$

where $\Phi(X) = \Theta^{-1}(X) = \mathrm{Diag}(XX^T)^{-\frac{1}{2}} XX^T \mathrm{Diag}(XX^T)^{-\frac{1}{2}}$. $\qquad\square$

*Proof of Lem. 5.2* . Using Eqs. (35) and (36), it is straightforward to see that

$$
\begin{aligned}
t \otimes P &= \mathrm{Exp}_{I_n} \left( t \mathrm{Log}_{I_n} (P) \right) \\
&= \mathrm{Exp}_{I_n} \left( (d_{I_n} \Theta)^{-1} \left( t \left( \Theta (P) - I_n \right) \right) \right) \\
&= \Theta^{-1} \left( (t\Theta (P) + (1 - t)I_n) \right) \\
&= \Phi \left( t \mathrm{Diag} \left( \mathrm{Chol} (P) \right)^{-1} \mathrm{Chol} (P) + (1 - t) I_n \right).
\end{aligned}
\tag{62}
$$

$\qquad\square$

*Proof of Thm. 5.3* . The gyroautomorphism can be determined from the binary operations (Ungar, 2005; 2012; 2014). Using Eq. (6), we can deduce that

$$
\begin{aligned}
\mathrm{gyr}_{ec}[P,Q]R &= \left( \ominus_{ec} \left( P \oplus_{ec} Q \right) \right) \oplus_{ec} \left( P \oplus_{ec} \left( Q \oplus_{ec} R \right) \right) \\
&\overset{(1)}{=} \left( \ominus_{ec} \left( \Theta^{-1} \left( \Theta (P) + \Theta (Q) - I_n \right) \right) \right) \oplus_{ec} \left( P \oplus_{ec} \left( \Theta^{-1} \left( \Theta (P) + \Theta (Q) - I_n \right) \right) \right) \\
&\overset{(2)}{=} \left( \Theta^{-1} \left( 3I_n - \Theta (P) - \Theta (Q) \right) \right) \oplus_{ec} \left( \Theta^{-1} \left( \Theta (P) + \Theta (Q) + \Theta (R) - 2I_n \right) \right) \\
&\overset{(3)}{=} \Theta^{-1} \left( 3I_n - \Theta (P) - \Theta (Q) + \Theta (P) + \Theta (Q) + \Theta (R) - 2I_n - I_n \right) \\
&= R.
\end{aligned}
\tag{63}
$$

The above derivation comes from the following.

(1) and (3) follow from Eq. (16).

(2) follows from Eq. (17).

We can deduce that $\mathrm{gyr}_{ec}[a,b] = I_d$. Next, we will prove that $(\mathrm{Cor}_n^+, \oplus_{ec})$ forms a gyrogroup, *i.e.*, it satisfies axioms G1, G2, G3, G4 and Gyrocommutative Law.

**Axiom (G1)**

*Proof.* For $P \in \mathrm{Cor}_n^+$, we have

$$
I_n \oplus_{ec} P = \Theta^{-1} \left( I_n + \Theta (P) - I_n \right) = P.
\tag{64}
$$

Therefore, $I_n$ is a left identity in $\mathrm{Cor}_n^+$. $\qquad\square$

**Axiom (G2)**

*Proof.* For $P \in \mathrm{Cor}_n^+$, let $Q = \Phi \left( 2I_n - \mathrm{Diag} \left( \mathrm{Chol} (P) \right)^{-1} \mathrm{Chol} (P) \right) = \Theta \left( 2I_n - \Theta (P) \right) \in \mathrm{Cor}_n^+$, we have

$$
\begin{aligned}
Q \oplus_{ec} P &= \Theta^{-1} \left( \Theta (Q) + \Theta (P) - I_n \right) \\
&= \Theta^{-1} \left( 2I_n - \Theta (P) + \Theta (P) - I_n \right) \\
&= I_n.
\end{aligned}
\tag{65}
$$

Therefore, $Q = \Theta \left( 2I_n - \Theta (P) \right)$ is a left inverse of $P$. $\qquad\square$

**Axiom (G3)**

*Proof.* For $P, Q, R \in \mathrm{Cor}_n^+$, we have

$$
\begin{aligned}
P \oplus_{ec} (Q \oplus_{ec} R) &= P \oplus_{ec} \left( \Theta^{-1} \left( \Theta(Q) + \Theta(R) - I_n \right) \right) \\
&= \Theta^{-1} \left( \Theta(P) + \Theta(Q) + \Theta(R) - 2I_n \right)
\end{aligned}
\tag{66}
$$

Since $\mathrm{gyr}_{ec}[P, Q]R = R$, we have

$$
\begin{aligned}
(P \oplus_{ec} Q) \oplus_{ec} \mathrm{gyr}_{ec}[P, Q]R &= (P \oplus_{ec} Q) \oplus_{ec} R \\
&= \left( \Theta^{-1} \left( \Theta(P) + \Theta(Q) - I_n \right) \right) \oplus_{ec} R \\
&= \Theta^{-1} \left( \Theta(P) + \Theta(Q) + \Theta(R) - 2I_n \right).
\end{aligned}
\tag{67}
$$

Therefore, $P \oplus_{ec} (Q \oplus_{ec} R) = (P \oplus_{ec} Q) \oplus_{ec} \mathrm{gyr}_{ec}[P, Q]R$. $\square$

**Axiom (G4)**

*Proof.* For $P, Q, R \in \mathrm{Cor}_n^+$, we have

$$
\begin{aligned}
&\mathrm{gyr}_{ec}[P \oplus_{ec} Q, Q]R \\
&= (\ominus_{ec} (P \oplus_{ec} Q \oplus_{ec} Q)) \oplus_{ec} ((P \oplus_{ec} Q) \oplus_{ec} (Q \oplus_{ec} R)) \\
&= \left( \ominus_{ec} \left( \Theta^{-1} \left( \Theta(P) + 2\Theta(Q) - 2I_n \right) \right) \right) \\
&\quad \oplus_{ec} \left( \left( \Theta^{-1} \left( \Theta(P) + \Theta(Q) - I_n \right) \right) \oplus_{ec} \left( \Theta^{-1} \left( \Theta(Q) + \Theta(R) - I_n \right) \right) \right) \\
&= \left( \Theta^{-1} \left( 4I_n - \Theta(P) - 2\Theta(Q) \right) \right) \oplus_{ec} \left( \Theta^{-1} \left( \Theta(P) + 2\Theta(Q) + \Theta(R) - 3I_n \right) \right) \\
&= \Theta^{-1} \left( 4I_n - \Theta(P) - 2\Theta(Q) + \Theta(P) + 2\Theta(Q) + \Theta(R) - 3I_n - I_n \right) \\
&= R.
\end{aligned}
\tag{68}
$$

Therefore, $\mathrm{gyr}_{ec}[P, Q]R = \mathrm{gyr}_{ec}[P \oplus_{ec} Q, Q]R$. $\square$

**Gyrocommutative Law**

*Proof.* Since we have proved that $\mathrm{gyr}_{ec}[P, Q] = I_d$, we have

$$
\begin{aligned}
\mathrm{gyr}_{ec}[P, Q](Q \oplus_{ec} P) &= Q \oplus_{ec} P \\
&= \Theta^{-1} \left( \Theta(Q) + \Theta(P) - I_n \right) \\
&= P \oplus_{ec} Q.
\end{aligned}
\tag{69}
$$

Therefore, it satisfies $P \oplus_{ec} Q = \mathrm{gyr}_{ec}[P, Q](Q \oplus_{ec} P)$. $\square$

Thus, $(\mathrm{Cor}_n^+, \oplus_{ec})$ forms a gyrogroup. Then, we will prove $(\mathrm{Cor}_n^+, \oplus_{ec}, \otimes_{ec})$ that endowed with the scalar multiplication, further forms a gyrovector space i.e., satisfying axioms V1, V2, V3, V4, V5 for gyrovector spaces.

**Axiom (V1)**

*Proof.* For $t \in \mathbb{R}$ and $P \in \mathrm{Cor}_n^+$, we have

$$
1 \otimes_{ec} P = \Theta^{-1} \left( \Theta(P) + (1 - 1) I_n \right) = P.
\tag{70}
$$

$$
0 \otimes_{ec} P = \Theta^{-1} (I_n) = I_n.
\tag{71}
$$

$$
t \otimes_{ec} I_n = \Theta^{-1} \left( tI_n + (1 - t) I_n \right) = I_n.
\tag{72}
$$

$$
-1 \otimes_{ec} P = \Theta^{-1} \left( -\Theta(P) + (1 - (-1)) I_n \right) = \ominus_{ec} P.
\tag{73}
$$

$\square$

**Axiom (V2)**

*Proof.* For $s, t \in \mathbb{R}$ and $P \in \mathrm{Cor}_n^+$, we have

$$(s + t) \otimes_{ec} P = \Theta^{-1} \left( (s + t) \Theta (P) + (1 - (s + t)) I_n \right). \tag{74}$$

$$
\begin{aligned}
& (s \otimes_{ec} P) \oplus_{ec} (t \otimes_{ec} P) \\
& = \left( \Theta^{-1} \left( s\Theta (P) + (1 - s) I_n \right) \right) \oplus_{ec} \left( \Theta^{-1} \left( t\Theta (P) + (1 - t) I_n \right) \right) \\
& = \Theta^{-1} \left( s\Theta (P) + (1 - s) I_n + t\Theta (P) + (1 - t) I_n - I_n \right) \\
& = \Theta^{-1} \left( (s + t) \Theta (P) + (1 - (s + t)) I_n \right).
\end{aligned} \tag{75}
$$

Therefore, $(s + t) \otimes_{ec} P = (s \otimes_{ec} P) \oplus_{ec} (t \otimes_{ec} P)$. $\qquad \square$

**Axiom (V3)**

*Proof.*
$$(st) \otimes_{ec} P = \Theta^{-1} \left( st\Theta (P) + (1 - st) I_n \right). \tag{76}$$

$$
\begin{aligned}
s \otimes_{ec} (t \otimes_{ec} P) & = s \otimes_{ec} \left( \Theta^{-1} \left( t\Theta (P) + (1 - t) I_n \right) \right) \\
& = \Theta^{-1} \left( s \left( t\Theta (P) + (1 - t) I_n \right) + (1 - s) I_n \right) \\
& = \Theta^{-1} \left( st\Theta (P) + (1 - st) I_n \right).
\end{aligned} \tag{77}
$$

Therefore, $(st) \otimes_{ec} P = s \otimes_{ec} (t \otimes_{ec} P)$. $\qquad \square$

**Axiom (V4)**

*Proof.* For $t \in \mathbb{R}$ and $P, Q, R \in \mathrm{Cor}_n^+$, since we have proved that $\mathrm{gyr}_{ec}[P, Q] = I_d$, we have

$$\mathrm{gyr}_{ec}[P, Q] (t \otimes_{ec} R) = t \otimes_{ec} R = t \otimes_{ec} \mathrm{gyr}_{ec}[P, Q]R. \tag{78}$$

$\qquad \square$

**Axiom (V5)**

*Proof.* For $s, t \in \mathbb{R}$ and $P, R \in \mathrm{Cor}_n^+$, we have

$$
\begin{aligned}
& \mathrm{gyr}_{ec}[s \otimes_{ec} P, t \otimes_{ec} P]R \\
& = \left( \ominus_{ec} \left( (s \otimes_{ec} P) \oplus_{ec} (t \otimes_{ec} P) \right) \right) \oplus_{ec} \left( (s \otimes_{ec} P) \oplus_{ec} (t \otimes_{ec} P \oplus_{ec} R) \right) \\
& = \left( \ominus_{ec} \left( (s + t) \otimes_{ec} P \right) \right) \\
& \quad \oplus_{ec} \left( \left( \Theta^{-1} \left( t\Theta (P) + (1 - t) I_n \right) \right) \oplus_{ec} \left( \Theta^{-1} \left( t\Theta (P) + (1 - t) I_n + \Theta (R) - I_n \right) \right) \right) \\
& = \left( \Theta^{-1} \left( 2I_n - (s + t) \Theta (P) - (1 - s - t) I_n \right) \right) \oplus_{ec} \left( \Theta^{-1} \left( (s + t) \Theta (P) - (s + t) I_n + \Theta (R) \right) \right) \\
& = \Theta^{-1} \left( 2I_n - (s + t) \Theta (P) - (1 - s - t) I_n + (s + t) \Theta (P) - (s + t) I_n + \Theta (R) - I_n \right) \\
& = \Theta^{-1} \left( \Theta (R) \right) \\
& = R.
\end{aligned} \tag{79}
$$

Therefore, $\mathrm{gyr}_{ec}[s \otimes_{ec} P, t \otimes_{ec} P]R = I_d$. $\qquad \square$

Thus, $(\mathrm{Cor}_n^+, \oplus_{ec}, \otimes_{ec})$ further forms a gyrovector space. $\qquad \square$

*Proof of Lem. 5.4* . Using the basic operations based on the LEC metric, we can deduce that

$$
\begin{aligned}
P \oplus_{lec} Q & = \mathrm{Exp}_P \left( \Gamma_{I_n \to P} \left( \mathrm{Log}_{I_n} (Q) \right) \right) \\
& = \mathrm{Exp}_P \left( (d_P (\log \circ \Theta))^{-1} \left( \log (\Theta (Q)) \right) \right). \\
& = \Theta^{-1} \circ \exp \left( \log (\Theta (P)) + \log (\Theta (Q)) \right) \\
& = \Phi \circ \exp \left( \log \left( \mathrm{Diag} \left( \mathrm{Chol} (P) \right)^{-1} \mathrm{Chol} (P) \right) + \log \left( \mathrm{Diag} \left( \mathrm{Chol} (Q) \right)^{-1} \mathrm{Chol} (Q) \right) \right),
\end{aligned} \tag{80}
$$

where $\Phi(X) = \Theta^{-1}(X) = \mathrm{Diag}(XX^T)^{-\frac{1}{2}} XX^T \mathrm{Diag}(XX^T)^{-\frac{1}{2}}$. $\qquad \square$

*Proof of Lem. 5.5* . Using Eqs. (39) and (40), it is straightforward to see that

$$
\begin{aligned}
t \otimes P &= \mathrm{Exp}_{I_n} \left( t \mathrm{Log}_{I_n} (P) \right) \\
&= \mathrm{Exp}_{I_n} \left( (d_{I_n} (\log \circ \Theta))^{-1} (t (\Theta (P))) \right) \\
&= \Theta^{-1} \circ \exp (t \log (\Theta (P))) \\
&= p^t.
\end{aligned}
\tag{81}
$$

$\square$

*Proof of Thm. 5.6* . The gyroautomorphism can be determined from the binary operations (Ungar, 2005; 2012; 2014). Using Eq. (6), we can deduce that

$$
\begin{aligned}
&\mathrm{gyr}_{lec}[P, Q]R \\
&= (\ominus_{lec} (P \oplus_{lec} Q)) \oplus_{lec} (P \oplus_{lec} (Q \oplus_{lec} R)) \\
&\overset{(1)}{=} \left( \ominus_{lec} \left( \Theta^{-1} \circ \exp (\log (\Theta (P)) + \log (\Theta (Q))) \right) \right) \\
&\quad \oplus_{lec} \left( P \oplus_{lec} \left( \Theta^{-1} \circ \exp (\log (\Theta (P)) + (\log (\Theta (P)))) \right) \right) \\
&\overset{(2)}{=} \Theta^{-1} \circ \exp \left( -\log (\Theta (P)) - \log (\Theta (Q)) + \log (\Theta (P)) + \log (\Theta (Q)) + \log (\Theta (R)) \right) \\
&= \Theta^{-1} \circ \exp (\log (\Theta (R))) \\
&= R.
\end{aligned}
\tag{82}
$$

The above derivation comes from the following.

(1) follows from Eq. (19).

(2) follows from Eq. (20).

We can deduce that $\mathrm{gyr}_{lec}[a, b] = I_n$. It is easy to verify axioms G1, G2, G3, G4. Thus, $(\mathrm{Cor}_n^+, \oplus_{lec})$ forms a gyrogroup. Endowed with the scalar multiplication $\otimes_{lec}$, it satisfies axioms V1, V2, V3, V4, V5 for EC gyrovector spaces(the proof follows similar logic as in Thm. 5.3). Therefore, $(\mathrm{Cor}_n^+, \oplus_{lec}, \otimes_{lec})$ further forms a gyrovector space. $\square$

