# OpenReview forum: "Algebraic SPD and Correlation Geometry: A Gyro Approach"
_ICLR.cc/2025/Conference — Submitted to ICLR 2025_

### Official Review · Reviewer_8a4t · 2024-10-28

**Soundness:** 2
**Presentation:** 2
**Contribution:** 2
**Rating:** 3
**Confidence:** 3

**Summary:**

This paper introduces a gyrovector space structure on symmetric positive definite (SPD) matrix manifolds based on the Power-Euclidean (PE) geometry and proposes two new gyro structures for full-rank correlation matrices using the Euclidean-Cholesky and Log-Euclidean-Cholesky metrics. These structures extend the applicability of deep neural networks in non-Euclidean geometry, with demonstrated effectiveness in knowledge graph completion tasks.

**Strengths:**

1. The paper introduces new gyrovector space structures for SPD and full-rank correlation matrix manifolds based on PE, EC, and LEC metrics, providing rigorous theoretical derivations that establish their mathematical validity and potential contribution to geometric deep learning.

2. The experiments validate the effectiveness of the proposed structures in knowledge graph completion tasks, demonstrating their applicability in handling non-Euclidean data.

**Weaknesses:**

1. From the perspective of theoretical construction, the paper **lacks significant theoretical innovation**, as the theoretical derivations are relatively simple and straightforward. The proofs in the appendix largely focus on verifying basic gyrovector space axioms, relying on direct calculations rather than innovative techniques. The approach primarily involves straightforward matrix operations, such as verifying identities and inverses through simple substitutions. The proofs for different metrics (PE, EC, LEC) follow nearly identical steps, showing limited structural novelty and repetitive methods across manifold types. Overall, the derivations lack deeper geometric insights or complex algebraic manipulation, limiting theoretical innovation.


2. The paper’s writing employs complex manifold language and notation, which may be inaccessible to machine learning researchers without a geometry background. Many concepts and derivations lack clear contextual explanations, making the content dense and challenging to follow. Additionally, the paper includes numerous technical details but does not provide sufficient simplifications or examples, making it hard for readers to grasp the main points quickly, potentially impacting readability and appeal.

3. While the experiments validate the proposed structures on knowledge graph completion, they are limited in scope and do not demonstrate performance on other practical tasks or non-Euclidean datasets. Moreover, the lack of publicly available code and the complex implementation details may hinder reproducibility. The experimental results focus on specific tasks and do not comprehensively showcase the method's potential advantages and limitations.

**Questions:**

1. The derivations mainly verify basic gyrovector space axioms through straightforward matrix operations, lacking complex geometric insights or novel algebraic techniques. So, what are the  theoretical innovations?

2. The complex manifold language and notation may be hard to follow for those without a geometry background. Is it possible to add simplified explanations, examples, or background to improve accessibility for a broader audience?

3. The experiments are limited to knowledge graph tasks, and the lack of public code and detailed implementation may hinder reproducibility. It is possible to expand the scope of applications and release the code for reproducibility?

---

> ### Author Response · Authors · 2024-11-25
> **Response to Reviewer 8a4t**
>
> $\textcolor{red}{\rm{Rebuttal}}$
>
> We thank the reviewer $\textcolor{green}{8a4t(R5)}$ for the instant reply. Below is our detailed response.
>
>
> **1.The theoretical innovations.**
>
> Previous works [a,b] have proposed three gyro-structures on SPD manifolds based on Affine-Invariant (AI), Log-Euclidean (LE) and Log-Cholesky (LC) metrics. The Power-Euclidean (PE) metric has demonstrated effectiveness in various applications. Considering its computational efficiency and its convergence to the LE metric as the matrix power approaches zero,  we propose a novel SPD gyro-structure under the PE metric. Furthermore, we are the first to introduce two gyro-structures on correlation matrix manifolds. The effectiveness of our approach is validated through extensive experiments on Knowledge Graph Completion (KGC) tasks.
>
> Furthermore, after performing the Cholesky decomposition of the correlation matrix, we observed that each row of the matrix belongs to an open hemisphere and exists an isometry between the open hemisphere and the Poincaré sphere. Building on this, we proposed a novel correlation matrix metric —the Poly Poincaré Cholesky metric (PPCM), and constructed a gyrovector space based on it. Due to time constraints, we conducted experiments on the WN18RR dataset, achieving remarkable results.
>
>
> Table I: Results on the WN18RR dataset.
>
> |Model| MRR| H@1| H@3| H@10|
> |-|-|-|-|-|
> |SPD$^R_{Sca}$ |41.7 | 36.5| 44.5| 51.1|
> |SPD$^{F1}_{Sca}$ | 40.8| 36.3| 42.9| 49.5|
> |SPD$^R_{Rot}$ |22.4| 8.4| 33.4| 47.3|
> |SPD$^{F1}_{Rot}$ |26.5| 18.1| 30.7| 42.9|
> |SPD$^R_{Ref}$ | 41.0| 37.1| 42.7| 47.6|
> |SPD$^{F1}_{Ref}$ |39.7| 35.9| 41.5| 46.3|
> |GyroGRLE-KGCNet | 41.5| 35.3| 44.9| 52.1|
> |**$GyroGRPPC-KGCNet$**| **44.7**| **40.3**| **46.8**| **52.7**|
>
> **3.Symbols and formulae.**
>
> In this paper, we propose several gyro-structures based on SPD and correlation matrices under different metrics. Due to the extensive group operations involved, a significant number of symbols and formulas are inevitably introduced. To address this, we will provide a more detailed explanation of the concept of gyrovector spaces and include a comprehensive review of the relevant literature in the introduction. Furthermore, given the dense content of the article, we will optimize the layout to present the material more clearly and make it easier to understand in the final version.
>
>
>
> **3.The reproducibility of code.**
>
> Our experimental code is modified based on [a], whose code is open source. The experimental results can be directly reproduced using the parameter descriptions and specific implementation techniques detailed in the experimental section of the main paper. Additionally, we plan to release our source code in the future.
>
>
>
>
> **References**
> > [a] Xuan Son Nguyen. The Gyro-Structure of Some Matrix Manifolds.
> >
> > [b] Xuan Son Nguyen. A Gyrovector Space Approach for Symmetric Positive Semi-definite Matrix Learning.

---

> > ### Comment · Reviewer_8a4t · 2024-11-26
> >
> > Thank you for your response and additional details. While I appreciate the effort to clarify your contributions, the theoretical innovations still seem incremental and lack deeper insights. Therefore, I would like to maintain my original score.

---

### Official Review · Reviewer_L3VE · 2024-11-04

**Soundness:** 3
**Presentation:** 2
**Contribution:** 2
**Rating:** 5
**Confidence:** 2

**Summary:**

This paper considers endowing the set of correlation matrices with a gyro (weak group) structure, which has the potentially of subsequently allowing deep learning over the set of correlation matrices. As highlighted by the authors, recent work over the past few years has suggested the potential gains from working in non-Euclidean geometry in certain tasks. Work has already been done on endowing the set of SPD matrices with a gryo structure.
More specifically:
- the authors propose a new gyro structure on the set of SPD matrices
- the authors propose two gyro structures for full-rank correlation matrices
- the performance of the DNN that they allow is evaluated on two datasets.

**Strengths:**

The paper is reasonably well written and seems to

**Weaknesses:**

Note: This reviewer is not an expert on gyrostructures, but I am familiar with some of the literature on DNN in non euclidean geometry.

-**Style:**
   -  The exposition might be improved by being a bit more specific. I understand the authors might want to provide just a brief overview of the literature, but some of the paragraphs become a bit vacuous (too many uses of the word "others", and extremely unspecific): "The space of SPD matrices forms a manifold, known as the SPD manifold which has been successfully applied in various fields. To respect the non-Euclidean geometry, several Riemannian structures on the SPD manifold were proposed .Due to the fast computation speed and theoretical convenience of the Power-Euclidean (PE) metric, and when the power tends to 0, this metric approaches the Log-Euclidean (LE) metric, building a bridge between Euclidean and LE metrics. Based on the above advantages, the PE metric has already seen successful applications in other fields."
  - I think the justification for this paper comes it much too late. The first few pages read more like an intellectual exercise, rather than something that could be useful. I am still a bit confused as to (a) why we need another gyrostructure on SPD matrices, and 2 new on correlation matrices: what is wrong with existing methods? Maybe, to make the exposition clearer, the authors could start with a use case example.

**Content:**
- Overall, my main comment is that the paper reads too much like a list (e.g. "here's 2 geometry and three metrics to consider). It does not provide insights as to (a) the current issues with the method (brief mention of the computational complexity only), or (b) try to make the reader understand why and when certain manifold and grystructure types are useful.
- in the experiments, it is unclear why the correlation matrix makes sense.

**Questions:**

See above

---

> ### Author Response · Authors · 2024-11-25
> **Response to Reviewer L3VE**
>
> $\textcolor{red}{\rm{Rebuttal}}$
>
> We thank Reviewer $\textcolor{red}{L3VE (R4)}$  for the constructive suggestions and insightful comments! In the following, we respond to the concerns in detail.
>
> **1.The presentation of the paper.**
>
> Thank you for your constructive suggestions. We will provide a more detailed overview of prior works in the introduction section, while clearly explaining the motivation behind our methods and highlighting its distinctions from existing approaches in the final version.
>
>
>
> **2. The current issues with the method.**
>
> In this paper, we propose a novel SPD gyro-structure under the Power-Euclidean (PE) metric and two gyro-structures on correlation matrix manifolds. Our experiments primarily focus on Knowledge Graph Completion (KGC) tasks. Using the proposed gyrovector space, we can also expand other Euclidean networks to manifolds, such as recurrent neural networks (RNN), multinomial logistic regression (MLR), fully connected (FC) and convolutional layers, etc. This is also the limitations of this paper.
>
>
> **3. The choice of manifold and gyro-structure types.**
>
> If the data exhibits inherent SPD properties (e.g., covariance matrices), the SPD manifolds should be utilized. Conversely, if the data captures statistical relationships (e.g., correlation matrices), the correlation matrix manifolds are more appropriate. The effectiveness of different gyro-structures may vary depending on the task or dataset. We hope that the proposed SPD gyro-structure under the PE geometry will be a viable alternative to existing Affine-Invariant (AI)-based or Log-Euclidean (LE)-based SPD gyro-structures when these gyro-structures demonstrate unsatisfying performance.
>
>
> **4. The significance of the full-rank correlation matrix.**
>
> In Section 4 of the main paper, we discuss the advantages of correlation matrices, which effectively capture more compact statistical information. However, since the metrics defined for SPD matrices cannot be directly applied to correlation matrices, we conducted an in-depth exploration and proposed two novel gyro-structures for full-rank correlation matrices under Euclidean-Cholesky (EC) and Log-Euclidean-Cholesky (LEC) metrics. Experimental results demonstrate that our method outperforms previous methods, further validating its effectiveness. Although our method performs slightly lower than the results under the PE metric, this highlights the suitability of different metrics for different tasks. Moreover, as shown in Table 6 in the main paper, we combine full-rank correlation matrices with SPD and Grassmannian manifolds, achieving superior performance.

---

### Official Review · Reviewer_djcf · 2024-11-10

**Soundness:** 2
**Presentation:** 3
**Contribution:** 2
**Rating:** 5
**Confidence:** 2

**Summary:**

The paper titled "Algebraic SPD and Correlation Geometry: A Gyro Approach" introduces a novel approach to leveraging gyro-structures on SPD manifolds, focusing on Power-Euclidean (PE) geometry and extending this to full-rank correlation matrix manifolds with Euclidean-Cholesky (EC) and Log-Euclidean-Cholesky (LEC) metrics. The main text is dense with theoretical formulations which are presented with rigorous mathematical proofs. Further, several experiments are conducted to validate the effectiveness of proposed methods on knowledge graph completion tasks.

**Strengths:**

1.The inclusion of detailed proofs and lemma/theorem statements provides a strong mathematical foundation and supports the credibility of the approach.

2.Using SPD and correlation matrix manifolds for knowledge graph completion is well-motivated and relevant, especially as these tasks require handling non-Euclidean data structures.

**Weaknesses:**

1.The application of gyro-structures on SPD manifolds and correlation matrices is indeed novel, but the paper does not clearly articulate the theoretical significance or unique advantages of using Power-Euclidean (PE) geometry over existing approaches like Affine-Invariant (AI) or Log-Euclidean (LE) methods. The work seems incremental without providing substantial theoretical or empirical evidence that PE geometry offers practical improvements beyond computational convenience. Especially, while gyro-structures are presented as an extension to non-Euclidean spaces, the paper does not establish a strong need or motivation for this approach within the broader context of machine learning or geometry-based learning. It lacks a thorough discussion on why gyro-structures would fundamentally enhance SPD or correlation matrix-based learning in a way that current methods do not.

2.Some key theoretical concepts and mathematical operations, such as those in gyrovector space theory and correlation matrix manifold construction, are highly technical and lack intuitive explanations. Additional clarification or simplified summaries would improve accessibility for readers unfamiliar with advanced Riemannian geometry.

3.On the experiments part, the related discussion lacks interpretive insights that would elucidate why the proposed gyro-structures outperform existing methods. In addition, while the paper compares its methods against SPD-based models and a few gyro-structure-based approaches, it lacks comparison with other state-of-the-art methods that might not rely on gyro-structures. This omission makes it unclear whether the proposed approach actually outperforms simpler or more commonly used techniques in manifold-based learning.

**Questions:**

Please refer to the weaknesses.

---

> ### Author Response · Authors · 2024-11-25
> **Response to Reviewer djcf**
>
> $\textcolor{red}{\rm{Rebuttal}}$
>
> We thank Reviewer $\textcolor{blue}{djcf(R3)}$ for the careful review and the suggestive comments. Below, we address the comments in detail.
>
> **1.Advantages of proposed methods.**
>
> [a] defined two gyros-tructures for SPD manifolds based on the Affine-Invariant (AI) and Log-Euclidean (LE) metrics. The Power-Euclidean (PE) metric has shown remarkable effectiveness across various fields. Given its computational efficiency and its convergence to the LE metric as the matrix power approaches zero, we further propose a gyrostructure based on the power Euclidean metric. Furthermore, we introduce, for the first time, two gyrostructures based on full-rank correlation matrices under the Euclidean-Cholesky (EC) and Log-Euclidean-Cholesky (LEC) metrics. Our experiments demonstrate that this approach outperforms existing AI and LE methods, validating the effectiveness of our proposed framework.
>
> Using gyrovector space, we extend fundamental operations from Euclidean space (such as matrix-matrix addition and scalar-matrix multiplication) to the manifolds, effectively generalizing Euclidean networks to manifolds. Since the core of the KGC task lies in constructing scoring functions, which inherently involve binary operations and matrix scaling (the key operations of gyrovector space), our approach is particularly well-suited for this task. Experimental results demonstrate that our proposed method outperforms existing approaches, further validating its effectiveness.
>
>
> **2.Symbols and formulae.**
>
> In this paper, we propose some gyro-structures for SPD and correlation matrices based on different metrics. Given the numerous group operations involved, the more symbols and formulas is unavoidable. To address this, we will provide more detailed explanations and strive to present the content in a clear and comprehensible manner in the final version.
>
>
>
> **3.Details of experiment.**
>
> In the KGC task, [b] first achieved significant results by learning entity embeddings in the Poincaré ball, demonstrating clear advantages over traditional Euclidean space. Then, [c] showed that SPD spaces provide superior geometric information representation. By embedding into SPD manifolds using Affine-Invariant (AI) metrics, further performance improvements were achieved. Subsequently, [a] proposed learning with the LE metric on SPD manifolds, combined with Grassmannian manifolds, leading to enhanced task performance. Therefore, we primarily compare our method with those proposed by [a] and [c], rather than methods based on Euclidean space. Experimental results further validate the effectiveness of our approach.
>
>
>
>
> **References**
>
> > [a] Xuan Son Nguyen. The Gyro-Structure of Some Matrix Manifolds.
> >
> > [b] Balazevic Ivana，Allen Carl，Hospedales, et al. Multi-relational poincaré graph embeddings.
> >
> > [c] López Federico，Pozzetti Beatrice, Trettel Steve, et al. Vector-valued distance and gyrocalculus on the space of symmetric positive definite matrices.

---

> > ### Comment · Reviewer_djcf · 2024-11-26
> > **Response to authors**
> >
> > Thank you for your response. While you've addressed some of my concerns, I still believe the novelty of this paper is marginal. Additionally, the authors have only compared it to a limited set of works, which is insufficient to demonstrate its broader relevance and usefulness.

---

### Official Review · Reviewer_KfSo · 2024-11-10

**Soundness:** 3
**Presentation:** 2
**Contribution:** 2
**Rating:** 3
**Confidence:** 3

**Summary:**

Gyrovector spaces extends transformations to hyperbolic geometry (originally), it captures operations analogous to addition and multiplication in Euclidean geometry. This paper studies gyrovector spaces for Symmetric Positive Definite (SPD) manifolds associated with Power-Euclidean metrics. Next, the paper looks into full-rank correlation matrices, as an example of SPD manifold, with two concrete Riemannian metrics: Euclidean-Cholesky and log-Euclidean-Cholesky metrics. The major contribution falls into proposing these metrics that can be applied to correlation matrices. The empirical study is implemented with the knowledge graph completion tasks, evaluating the proposed metrics on two datasets. The performance is marginally improved over previous work.

**Strengths:**

In general, the paper is well-organized. It is still notation-heavy due to the group operations etc. but I see the authors put efforts in making things clear.

To the best of my knowledge the paper is mathematically solid. I only had time to check the proof in section 3. I am happy to check more details if anything comes up, but at this moment I believe the later proofs follow quite similarly from the definitions.

It is illuminating to discuss the geometry of correlation matrices separately from the general SPD matrices, with a good motivation presented in section 4.

**Weaknesses:**

The presentation can still be improved.
- PE geometry has been emphasized quite a lot but not really explained what it is (for general audience). The only relevant description I could find locates in line 169, which is not a clear definition.
- I would say the related literature has been covered quite well in the introduction section. Maybe, it would be better to mention more motivation, technique used and other discussions in the intro, and have a related work section. It could be helpful to discuss various Riemannian geometries and how (and if) gyrovector spaces are different on each.
- I like section 4. It might be regarded as discruptive, because we are in the middle of technical sections but suddenly something more like prelimianry appears. I am personally fine with what it is, it should be also fine to cut it shorter, and merge with section 5
- Maybe this is the most concerned comment. The method for KGC task is compressed a bit too much. I understand it is not very complicated and extends from prior work. But for readers, the KGC is a new mateiral and it is better to introduce your method with more intuitions before having heavy notations.

My major concern is two-fold: the technical novelty is limited and the experiment results are not convincing enough.

The mathematical elements, though non-trivial, heavily inherit from the gyro group definitions. The gyro structure has been stuided for other Riemannian metrics, at this point, I do not see what is the techincal barrier to extend these definitions to another metric.

The empirical study has three issues, with importance in order: (1) The improvement is very small, especially, there is not a significant gap from the previous Gyro-LE metric. (2) The baselines are not inclusive -- only a similar prior work and rather trivial SPD transformations. (3) Two datasets are not enough.

**Questions:**

Just to make sure I did not miss it: what is the new technical challenge to extend the gyro structure to PE (and two others for correlation matrices) geometry? Especially, comparing to the prior work.

The EC and LEC metrics are tailored to benefit correlation matrices better, but this is not corroborated by table 2 and 3. Results by PE seems to be better mostly. Do we have any understanding on this?

Are there other tasks to be considered fit to evaluate the proposed metrics? Deep learning can in fact add more randomness into evaluation, is it possible to evaluate on tasks that are more dependent on "distance between matrices" itself? I am thinking of clustering and some time series analysis tasks such as anomaly detection or change point detection, only for reference.

---

> ### Author Response · Authors · 2024-11-25
> **Response to Reviewer KfSo**
>
> $\textcolor{red}{\rm{Rebuttal}}$
>
> We appreciate the constructive comments from Reviewer $\textcolor{purple}{KfSo(R2)}$. The following is our detailed response.
>
>
> **1.Symbol and presentation.**
>
> Thank you for your very constructive suggestions! To clarify, we will make corrections based on the identified issues  in the final version. We will expand the introduction section to introduce more motivations for our method and highlight its distinctions from previous approaches. Furthermore,  we will merge Section 4 with Section 5 and provide more comprehensive details on the KGC task.
>
>
> **2.Difference from prior work.**
>
> Previous works [a,b] have proposed three gyro-structures on SPD manifolds based on Affine-Invariant (AI), Log-Euclidean (LE) and Log-Cholesky (LC) metrics. The Power-Euclidean (PE) metric has demonstrated effectiveness in various applications. Considering its computational efficiency and its convergence to the LE metric as the matrix power approaches zero,  we propose a novel SPD gyro-structure under the PE metric. Furthermore, we are the first to introduce two gyro-structures on correlation matrix manifolds. The effectiveness of our approach is validated through extensive experiments on Knowledge Graph Completion (KGC) tasks.
>
> **3.Experimental performance on  full-rank correlation matrices.**
>
> we propose two novel gyro structures for full-rank correlation matrices,induced by the theoretically and computationally efficient Euclidean-Cholesky (EC) and Log-Euclidean-Cholesky (LEC) metrics. Although our method performs slightly below the results under the proposed PE metric in Tables 2 and 3 of the main paper, it still outperforms previous methods, demonstrating its effectiveness while highlighting that different metrics are suited to different tasks. Furthermore, in Table 6, we combine full-rank correlation matrices with SPD and Grassmannian manifolds, achieving superior results.
>
>
> **4.Application to other tasks.**
>
> Thank you for your valuable suggestions. In the KGC task, distance calculation plays a pivotal role, particularly in constructing the scoring function. Our experimental results demonstrate the effectiveness of the proposed framework. We greatly appreciate your suggestion to explore tasks such as clustering, anomaly detection, and change point detection, which rely heavily on distance metrics. These are insightful directions, and we will consider them in future work to further validate the versatility and robustness of our proposed indicators.
>
>
> **References**
> > [a] Xuan Son Nguyen. The Gyro-Structure of Some Matrix Manifolds.
> >
> > [b] Xuan Son Nguyen. A Gyrovector Space Approach for Symmetric Positive Semi-definite Matrix Learning.

---

### Official Review · Reviewer_uQXC · 2024-11-15

**Soundness:** 3
**Presentation:** 2
**Contribution:** 2
**Rating:** 5
**Confidence:** 3

**Summary:**

This paper presents novel algebraic structures (gyro-structures) for Symmetric Positive Definite (SPD) manifolds and full-rank correlation matrices, inspired by the extension of Deep Neural Networks (DNNs) to non-Euclidean geometries. Leveraging Power-Euclidean (PE) geometry, the authors introduce gyro-structures for SPD manifolds, alongside Euclidean-Cholesky (EC) and log-Euclidean-Cholesky (LEC) metrics for correlation matrices. Empirical validation is conducted on knowledge graph completion tasks, showing improvements in computational efficiency and model accuracy.

**Strengths:**

**Novelty:** Introduces new gyro-structures tailored for SPD manifolds using PE geometry, which recovers existing LE spaces and provides flexibility.

**Theoretical Rigor:** Detailed theoretical results, including binary operations and scalar multiplication, rigorously define the gyro-structures.

**Empirical Validation:** Experiments on knowledge graph tasks demonstrate that the proposed gyro-structures can outperform existing models in some metrics.

**Weaknesses:**

The most critical weakness is the lack of clear explanation of how the theoretical concepts are actually implemented in the experimental algorithms:
- Missing details on algorithm implementation: The experimental section typically needs to elaborate on the details of the algorithm's implementation, including framework overview, pseudo-code, parameter selection, and specific implementation techniques. Without these details, the practical applicability of the theoretical results may be questioned.
- Insufficient correspondence between experimental validation and theoretical support: Theoretical research should directly support the design and analysis of experiments, but the connection between the experiments presented and the preceding theoretical derivations is loose in this paper. It would be better to add an explanation of how theory inspires the design of algorithm.

**Questions:**

How do the gyro-structures affect performance in other DNN applications, such as image or text classification?

---

> ### Author Response · Authors · 2024-11-25
> **Response to Reviewer uQXC**
>
> $\textcolor{red}{\rm{Rebuttal}}$
>
> We thank the reviewer $\textcolor{orange}{uQXC(R1)}$ for the valuable comments. Our detailed responses are given below.
>
> **1. Details on the algorithm.**
>
> The core of the  Knowledge Graph Completion (KGC) task lies in the definition of the scoring function, which we have presented in Section 6.1 of the main paper, specifically in Equation 22. The binary operations and matrix scaling are key operations in the proposed gyrovector space. Details regarding parameter selection, initialization, and specific implementation techniques are further elaborated in Section 6.2. Additionally, we plan to release our source code soon.
>
>
> **2. Correspondence between experimental validation and theoretical support.**
>
> In the task of KGC, [a] was the first to achieve significant results by learning entity embeddings in the Poincaré ball, demonstrating clear advantages over traditional Euclidean spaces. Building on this, [b] found Symmetric Positive Definite (SPD) spaces offer superior geometric information representation. By embedding into SPD manifolds based on Affine-Invariant (AI) metrics, further improvements were achieved. Subsequently, [c] proposed learning with the  Log-Euclidean (LE) metric on SPD manifolds, combined with Grassmannian manifolds, achieving enhanced performance.
>
> Building on these advancements, we propose a novel approach to learning on SPD manifolds under the Power-Euclidean (PE) metric and also introduce learning with Euclidean-Cholesky(EC) and Log-Euclidean-Cholesky (LEC) metrics on full-rank correlation matrix manifolds, combined with Grassmannian manifolds to achieve superior results. Since the score function of KGC involves matrix addition and matrix scaling operations, which are the key operations of our proposed gyrovector space, the KGC task has important theoretical relevance to our method.
>
>
> **3. Application of the proposed method in other fields.**
>
> In this paper, we propose a novel SPD gyro-structure under the PE metric and two new gyro-structures for full-rank correlation matrices under the Euclidean-Cholesky (EC) metric and the Log-Euclidean-Cholesky (LEC) metrics. In the KGC task, we apply our theoretical framework to construct a scoring function, achieving effective results. Furthermore, the proposed theory has the potential to extend other Euclidean networks to manifolds, enabling applications in tasks such as text or image classification.  At the same time, this is also the limitation of this paper and a direction for future research.
>
>
> **References**
>
> > [a] Balazevic Ivana，Allen Carl，Hospedales, et al. Multi-relational poincaré graph embeddings.
> >
> > [b] López Federico，Pozzetti Beatrice, Trettel Steve, et al. Vector-valued distance and gyrocalculus on the space of symmetric positive definite matrices.
> >
> > [c] Xuan Son Nguyen. The Gyro-Structure of Some Matrix Manifolds.

---

### Meta-Review · Area_Chair_3ZjQ · 2024-12-17

**Metareview:**

The paper introduces a gyrovector space structure on the manifold of symmetric positive definite (SPD) matrices based on the Power-Euclidean (PE) geometry and proposes two new gyro structures for full-rank correlation matrices using the Euclidean-Cholesky and Log-Euclidean-Cholesky metrics. Numerical experiments are provided to illustrate the mathematical framework.

While reviewers agree that the proposed framework is new, they also raise concerns about its theoretical innovation and motivation.
It is not clear why there is a need for another gyrovector space structure on the SPD manifold and which problems it can tackle that cannot already be solved by existing methods. The theoretical derivations are straightforward verification of the axioms of gyrovector space.The numerical experiments are limited and do not yet provide compelling evidence on the effectiveness of the proposed method.

**Additional Comments On Reviewer Discussion:**

Main points raised by reviewers

- Lack of theoretical novelty, innovation: Reviewers KfSo,  djcf, L3VE,  8a4t.

- Limited experimental validation: Reviewers KfSo,  djcf, 8a4t

In response, the authors clarified the difference of their proposed method compared with previous work, the significance of correlation matrices, and provided some new experimental results. However, reviewers remained convinced that the contributions are marginal.

These points together lead to the reject decision of the paper.

---

### Decision · Program_Chairs · 2025-01-22

Reject